# Glucose hypometabolism prompts RAN translation and exacerbates C9orf72-related ALS/FTD phenotypes

Andrew T Nelson[1,4], Maria Elena Cicardi [ID][1,4], Shashirekha S Markandaiah[1], John YS Han[2], Nancy J Philp [ID][2], Emily Welebob[1], Aaron R Haeusler [ID][1], Piera Pasinelli[1], Giovanni Manfredi [ID][3], Hibiki Kawamata [ID][3] & Davide Trotti [ID][1 ✉]

## Abstract

**The most prevalent genetic cause of both amyotrophic lateral sclerosis and frontotemporal dementia is a (GGGGCC)$_n$ nucleotide repeat expansion (NRE) occurring in the first intron of the *C9orf72* gene (C9). Brain glucose hypometabolism is consistently observed in C9-NRE carriers, even at pre-symptomatic stages, but its role in disease pathogenesis is unknown. Here, we show alterations in glucose metabolic pathways and ATP levels in the brains of asymptomatic C9-BAC mice. We find that, through activation of the GCN2 kinase, glucose hypometabolism drives the production of dipeptide repeat proteins (DPRs), impairs the survival of C9 patient-derived neurons, and triggers motor dysfunction in C9-BAC mice. We also show that one of the arginine-rich DPRs (PR) could directly contribute to glucose metabolism and metabolic stress by inhibiting glucose uptake in neurons. Our findings provide a potential mechanistic link between energy imbalances and C9-ALS/FTD pathogenesis and suggest a feedforward loop model with potential opportunities for therapeutic intervention.**

**Keywords** ALS; C9orf72; FTD; Glucose Hypometabolism; RAN Translation
**Subject Categories** Metabolism; Molecular Biology of Disease; Translation & Protein Quality

## Introduction

Amyotrophic lateral sclerosis (ALS) is a progressive adult-onset neurodegenerative disease characterized by the loss of upper and lower motor neurons, leading to muscle wasting, paralysis, and eventually death (Hardiman et al, 2017). It is a highly hetero-geneous disease with numerous known genetic causes (Hardiman

et al, 2017). The most common of these is a -(G$_4$C$_2$)$_n$- nucleotide repeat expansion (NRE) in a non-coding region of the *C9orf72* gene (C9), which accounts for up to 20% of all ALS cases, as well as a subset of frontotemporal dementia (FTD) cases (DeJesus-Hernandez et al, 2011; Renton et al, 2011).

Common features of various forms of ALS and other neurodegenerative diseases include altered metabolism and bioenergetic homeostasis (Cunnane et al, 2020; Nelson and Trotti, 2022). Specifically, ALS patients often exhibit systemic hypermetabolism, mitochondrial dysfunction, and brain glucose hypometabolism (De Vocht et al, 2020; Fayemendy et al, 2021; Popuri et al, 2021; Smith et al, 2019). Targeting these alterations through various metabolic treatments and dietary manipulations has shown promise as a potential therapeutic strategy (Beghi et al, 2013; Ludolph et al, 2020). Furthermore, in C9-NRE carriers, metabolic alterations—especially glucose hypometabolism—can occur many years before symptom onset (Popuri et al, 2021; Xia et al, 2023). Therefore, energy imbalance is an early manifestation in C9-NRE carriers. However, whether it plays a role in disease pathogenesis and progression remains unknown.

A well-established pathogenic mechanism associated with C9-ALS/FTD is the aberrant production of dipeptide repeats (DPRs) through repeat-associated non-ATG (RAN) translation. Due to bidirectional transcription of the NRE, five distinct DPRs are produced, including poly-glycine-alanine (GA) and poly-glycine-arginine (GR) from the sense (G$_4$C$_2$) transcript; poly-proline-alanine (PA) and poly-proline-arginine (PR) from the antisense (C$_4$G$_2$) transcript; and poly-glycine-proline (GP) from both transcripts (Ash Peter et al, 2013; Mori et al, 2013; Zu et al, 2013). Accumulation of DPRs in aggregated forms has been detected in the central nervous system (CNS) of ALS/FTD patients (Ash Peter et al, 2013; Mori et al, 2013). Moreover, the GA, GR, and PR species are toxic in both cultured neurons and in vivo (Hao et al, 2019; Jensen et al, 2020; Kwon et al, 2014; May et al, 2014; Verdone et al, 2022; Wen et al, 2014). For these reasons, DPR production and accumulation are thought to contribute to neurodegeneration in C9 patients directly. Still, a complete

[1]Weinberg ALS Center, Vickie and Jack Farber Institute for Neuroscience, Department of Neuroscience, Thomas Jefferson University, Philadelphia, PA 19107, USA. [2]Department of Pathology and Genomic Medicine, Thomas Jefferson University, Philadelphia, PA 19107, USA. [3]Feil Family Brain and Mind Research Institute, Weill Cornell Medicine, 407 East 61st Street, New York, NY 10065, USA. [4]These authors contributed equally: Andrew T Nelson, Maria Elena Cicardi. ✉E-mail: davide.trotti@jefferson.edu

understanding of the RAN translation mechanism—including how it can be triggered or modified by other disease-related events (such as energy imbalance)—is lacking.

Remarkably, several brain regions that are hypometabolic in C9-NRE carriers—including the frontal and temporal cortices—also tend to be rich in DPR pathology (Ash Peter et al, 2013; De Vocht et al, 2020; Popuri et al, 2021; Schludi et al, 2015), which raises the possibility that glucose hypometabolism and DPR formation are mechanistically linked. Notably, cellular stress can selectively enhance RAN translation through activation of the integrated stress response (ISR) (Cheng et al, 2018; Green et al, 2017; Westergard et al, 2019), and glucose deprivation is a well-established activator of the ISR (Pakos-Zebrucka et al, 2016; Yang et al, 2000; Yang et al, 2013). Therefore, we hypothesized that glucose hypometabolism could enhance DPR production in C9-ALS/FTD. However, the reverse hypothesis that DPRs can contribute to glucose hypometabolism is also worth considering. Notably, pathogenic proteins associated with several other neurodegenerative diseases (including β-amyloid and polyglutamine-expanded Huntingtin, linked to Alzheimer's and Huntington's disease, respectively) can interfere with neuronal glucose uptake (Li et al, 2012; Prapong et al, 2002). Therefore, we also propose that the C9-linked DPRs could act similarly and contribute to or exacerbate metabolic stress.

In the present study, we investigated the brain metabolite profile of asymptomatic BAC transgenic mice carrying the C9orf72-linked repeat expansion (C9-BAC). We identified significant alterations in glucose metabolic pathways and impairment in steady-state brain ATP levels. We then demonstrated that glucose hypometabolism promotes DPR production and impacts the typical survival profile of cultured neurons expressing the C9-NRE, to a certain extent, *via* activation of the GCN2 arm of the ISR. We also found that chronic exposure of the C9-BAC mouse to glucose hypometabolism exacerbates the production of DPRs and causes motor dysfunction. Finally, we show that PR can impair neuronal glucose uptake, reduce metabolic flux, and activate the ISR. Our results support a feedforward loop linking glucose hypometabolism, cellular stress, and DPR formation, opening several therapeutic intervention opportunities.

## Results

### Energy metabolites are altered in the C9orf72 BAC mouse brain

Since glucose is classically considered the primary energy substrate in the brain (Dienel, 2019), we hypothesized that reductions in brain glucose uptake (as seen in C9orf72-ALS/FTD patients (De Vocht et al, 2020; Popuri et al, 2021)) would be accompanied by altered brain energy production. Ideally, this would be assessed by measuring metabolite levels directly in patient postmortem tissue. However, this is unfeasible, mainly due to metabolite degradation during the postmortem interval (Chighine et al, 2021). Therefore, we leveraged a C9-BAC transgenic mouse model, which carries the entire human C9orf72 gene with a [GGGGCC] repeat expansion (Fig. EV1A) and displays C9orf72-related pathology (i.e., RNA foci and DPR inclusions) but otherwise lacks overt behavioral phenotypes (O'Rourke et al, 2015). We performed targeted liquid chromatography-mass spectrometry (LC-MS)-based metabolite profiling to measure the abundance of >200 different metabolites in the frontal cortex of both C9-BAC animals and wild-type littermate controls (Fig. 1A; Dataset EV1). Principal component analysis (PCA) indicated that the relative differences in this metabolite panel were sufficient in separating C9-BAC from wild-type (Fig. 1B). Remarkably, out of all metabolites analyzed, ATP was the most significantly altered and was decreased by ~2-fold in C9-BAC brain (frontal cortex) relative to wild-type; ATP:ADP and ATP:AMP ratios were also significantly reduced (Fig. 1C; Dataset EV1). Such an ATP imbalance could be explained by dysfunction of the two main energy-producing metabolic pathways: (1) glycolysis and (2) oxidative phosphorylation (Dienel, 2019). Interestingly, we observed alterations to both of these pathways in the brains of C9-BAC animals. Enrichment analysis revealed that glycolysis was significantly enriched among metabolic pathways and decreased in the brain of C9-BAC (Fig. 1D). Upon closer analysis, despite no differences in brain glucose levels (Fig. EV1B), we observed significant alterations in the glycolytic intermediates glucose-6-phosphate (G6P), glyceraldehyde-3-phosphate (GADP), and phosphoenolpyruvate (PEP) in the frontal cortex of the C9-BAC mice (Fig. 1E), which indicates that glycolytic function may be altered. We also observed an enrichment of citric acid cycle metabolites (decreased in C9; Fig. 1D) and a decrease specifically in α-ketoglutarate, a rate-limiting intermediate of the citric acid cycle and precursor for several amino acids, including glutamate (Wu et al, 2016) (Fig. EV1C). Interestingly, C9-BAC animals also showed *higher* levels of several amino acids, including lysine, isoleucine, cysteine, and ornithine (Fig. EV1D), as well as significant enrichment of metabolite sets involved in amino acid metabolism (Fig. 1D). This could reflect a compensatory shift towards increased metabolism or transport of amino acids for energy production. Finally, we detected significant decreases in the key redox cofactors $NAD^+$ and $NADP^+$ in C9-BAC animals with no changes in their reduced counterparts (NADH and NADPH, respectively; Figs. 1F and EV1E), indicating a hyper-reduced redox state, which could contribute to glycolytic inefficiency by limiting the pool of available $NAD^+$. We also detected a decrease in the levels of nicotinamide mononucleotide (NMN; Fig. 1F), an essential precursor of $NAD^+$ (Xie et al, 2020), suggesting a potential defect in $NAD^+$ biosynthesis. Overall, our metabolomics data indicate that the C9orf72 repeat expansion mutation is sufficient to alter ATP levels and the $NAD^+$/NADH ratio in the mouse frontal cortex.

### Glucose hypometabolism drives dipeptide repeat accumulation by enhancing RAN translation

We next evaluated the potential impact of energy imbalance on a potential pathogenic mechanism of C9-ALS/FTD: the production of DPRs by RAN translation of the C9-NRE (Mori et al, 2013; Zu et al, 2013). Given our observations of glycolytic imbalance in the frontal cortex of the C9-BAC mouse, as well as the known overlap between the brain distribution of DPR-positive inclusions (Ash Peter et al, 2013; Schludi et al, 2015) and brain glucose hypometabolism in human mutation carriers (De Vocht et al, 2020; Popuri et al, 2021), we examined the role of glucose hypometabolism as a modifier of RAN translation. To reliably and reproducibly model RAN translation in vitro, we used a human synapsin-driven lentiviral vector to ectopically express a

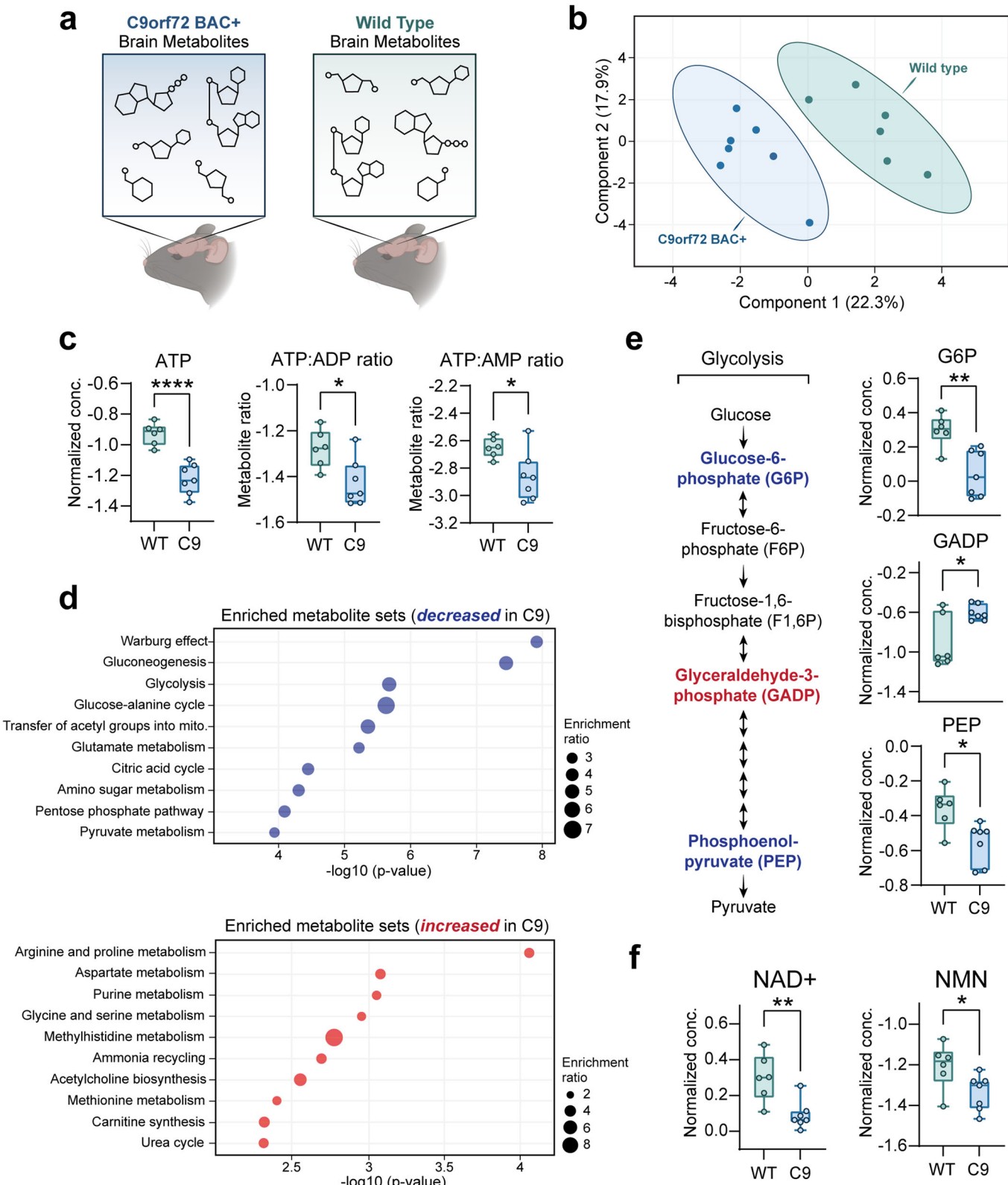

(GGGGCC)$_{188}$ repeat expansion with a 3' GFP tag (lacking an ATG start codon) in primary rodent neurons (Fig. 2A). Notably, we included the region of the C9orf72 gene located immediately upstream of the G$_4$C$_2$ repeat expansion—including exon 1a and a portion of intron 1—which is thought to be required to initiate RAN translation from the sense strand (Tabet et al, 2018). Indeed, neurons transduced with this vector exhibit a clear GFP signal that accumulates into perinuclear and dendritic DPR inclusions

**Figure 1.   The C9orf72-linked G₄C₂ repeat expansion disrupts brain energy balance.**

Impact of C9orf72-linked $G_4C_2$ repeat expansion on brain energy balance. (A) Schematic depicting brain metabolite profiling experimental setup. (B) Principal component analysis (PCA) of the complete metabolomics panel. (C) Liquid chromatography-mass spectrometry (LC-MS) measurement of relative ATP concentrations, ATP:ADP ratios, and ATP:AMP ratios in C9-BAC animals versus WT animals. (D) Enrichment analysis of metabolite sets in C9-BAC animals versus WT animals. Enrichment ratio represents the number of metabolites within each metabolite set that are either increased (in blue) or decreased (in red) in the frontal cortex of C9-BAC versus WT animals. (E) Schematic of glycolysis pathway, with significantly altered glycolytic intermediates highlighted in blue text (decreased in C9-BAC animals) or red text (increased in C9-BAC animals). LC-MS measurement of relative concentrations of glucose-6-phosphate (G6P), glyceraldehyde-3-phosphate (GADP), and phosphoenolpyruvate (PEP) in C9-BAC versus WT animals. (F) LC-MS measurement of nicotinamide adenine dinucleotide (NAD⁺) and nicotinamide mononucleotide (NMN) concentrations in C9-BAC versus WT animals. All individual metabolite data are shown as median-normalized and log-transformed values (abbreviated as "Normalized conc."). For all data, $n = 7$ C9orf72 BAC+ (C9-BAC) and 6 littermate control wild type (WT) animals. For all box and whisker plots, box edges denote upper and lower quartiles, horizontal lines within each box denote median values, whiskers denote maximum and minimum values, and shaded circles denote individual values for each animal. Student's two-tailed t-test, *$p < 0.05$; **$p < 0.01$; ****$p < 0.0001$.

(Fig. 2B), which are detectable by Western blot using both GFP- and DPR-specific antibodies (Fig. 2C). We then established an in vitro glucose hypometabolism paradigm by replacing a portion (up to 40%) of the glucose in the media with the glycolysis inhibitor 2-deoxyglucose (2DG), which caused downregulation of energy metabolism gene ontology (GO) pathways at the transcriptional level (Fig. EV2A). We also found that 2DG treatment caused a sharp reduction of both extracellular acidification rate (ECAR) and oxygen consumption rate (OCR), as measured by the Seahorse extracellular flux assay (Fig. EV2B), indicating functional impairment of energy metabolism. We then applied the 2DG treatment paradigm to neurons transduced with the RAN translation vector (Fig. 2D). Remarkably, we found that DPR formation was highly 2DG dose-dependent, with neurons treated with the highest 2DG concentration (10 mM) accumulating ~six times the number of DPR inclusions compared to non-treated cells (Fig. 2E), strongly suggesting that glucose hypometabolism increases DPR accumulation through RAN-T but not through increased transcriptional activity (Fig. 2F).

As a control, we performed the same experiment, but instead of using the RAN translation vector to drive DPR production, we used a separate ATG-driven vector encoding 50 codon-optimized GA repeats with a 3' GFP tag (Fig. 2G). Importantly, the 2DG treatment did not affect the abundance of GA aggregates from this ATG-driven vector (Fig. 2H). This indicates that glucose hypometabolism selectively enhances ran translation while having limited effects on ATG-driven translation. The findings suggest that RAN-translated DPR accumulation is due to increased production rather than decreased turnover since the GA aggregates made from both vectors are subject to the same molecular chaperones and degradative mechanisms. Finally, we observed no impact from the 2DG treatment on GFP mRNA levels (Fig. 2I), which indicates that the increase in aggregates is not due to changes in viral transduction or transcription efficiencies but reflects the upregulation of RAN translation.

## Glucose hypometabolism causes DPR accumulation through the GCN2 arm of the ISR

To elucidate the mechanism through which glucose hypometabolism enhances the accumulation of DPRs, we first used RNA sequencing to identify the significant transcriptomic changes that occur in C9orf72 patient-derived induced pluripotent stem cell (iPSC) neurons upon 2DG treatment. Among the most highly upregulated pathways were those implicated in stress response, including "response to the endoplasmic reticulum (ER) stress,"

"amino acid transport," and "amino acid biosynthesis" (Fig. 3A), which led us to consider cellular stress signaling as a potential mediator of increased RAN translation. In line with this, several studies have demonstrated that cellular stress can increase RAN translation by activating the ISR (Cheng et al, 2018; Green et al, 2017; Westergard et al, 2019). The key on-switch for the ISR is phosphorylation of eIF2α, which is carried out by four different kinases—HRI, PKR, PERK, and GCN2—each activated by different forms of stress (Pakos-Zebrucka et al, 2016). A major outcome of eIF2α phosphorylation is ATF4 translational activation. ATF4 then translocates to the nucleus and activates transcription of numerous target genes (Pakos-Zebrucka et al, 2016). Indeed, in our RNA-seq dataset, we observed upregulation of ATF4 and multiple known targets of ATF4 (Fig. 3B), indicating that 2DG treatment activates the ISR in C9orf72 patient-derived iPSC neurons. We also identified upregulation of ATF4 in spinal cord lysates from C9orf72 patients (Fig. EV3A), highlighting the relevance of ISR activation in C9-ALS/FTD.

We then confirmed that 2DG treatment activates the ISR in our primary rodent neuron model. Using immunofluorescent staining, we observed a 2DG dose-dependent increase in ATF4 nuclear intensity, specifically in MAP2-positive neurons (Fig. 3C), which strongly correlated with DPR accumulation ($R^2 > 0.8$) (Fig. 3D). In the same neurons we also measured by western blot an approximately 60% increase in the phospho/dephospho eIF2α ratio (Fig. 3E). These combined lines of evidence suggested that the increase in DPR accumulation was indeed mediated by ISR activation. To provide more evidence, we tested whether direct pharmacological inhibition of the ISR could also inhibit DPR accumulation caused by 2DG. Specifically, we blocked GCN2 because it is activated by nutrient deprivation stress (Fig. 4A). Remarkably, we found that treatment of the neuronal cultures with A92—a pharmacological inhibitor of GCN2—completely prevented the increase in DPR accumulation caused by 2DG (Fig. 4B,C) with an $IC_{50}$ value of 1.6 μM (Fig. 4d). Importantly, A92 had no deleterious effect on neuronal survival (Fig. EV3B). Furthermore, ISRIB (a small molecule inhibitor of the ISR) could also block the increase in DPR formation caused by 2DG (Fig. EV3C), further confirming the involvement of the ISR in augmenting DPR production. Lastly, we demonstrated that inhibition of GCN2 with A92 can block DPR formation caused by 2DG in human control iPSC-derived neurons transduced with our RAN translation reporter construct (Fig. EV3D). Together, our data indicate that glucose hypometabolism enhances neuronal RAN translation through activation of the GCN2 arm of the ISR.

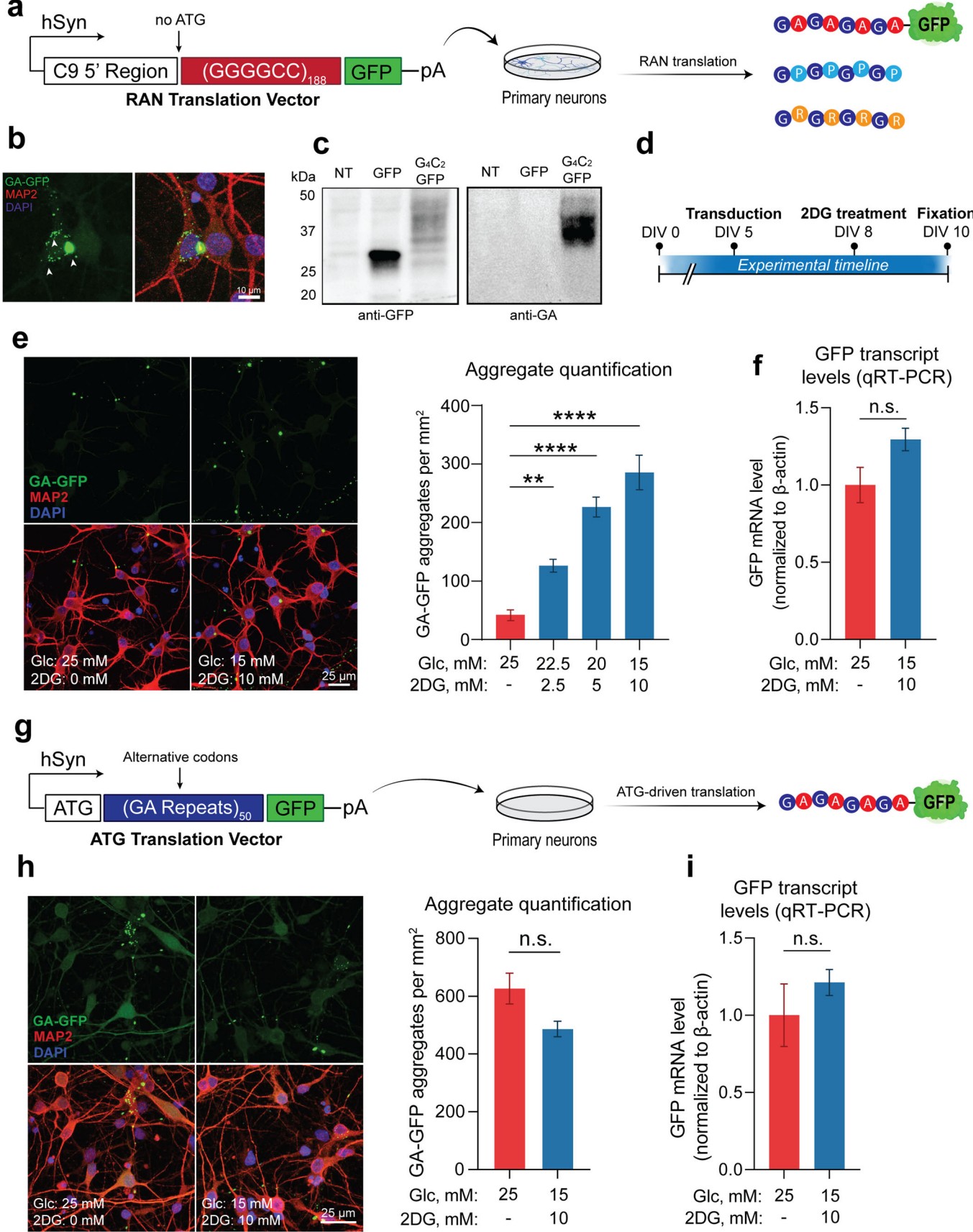

**Figure 2. Glucose hypometabolism triggers accumulation of DPRs.**

Cortical neurons transduced with $(G_4C_2)_{188}$ repeats are exposed to hypometabolic conditions. (A) Schematic of lentiviral RAN translation vector used for the experiments, containing a $(G_4C_2)_{188}$ repeat expansion with the 5′ flanking region of the C9orf72 gene (including exon 1a and intron 1) and a downstream GFP tag lacking an ATG start codon in frame with GA. The entire construct was driven by the human synapsin (hSyn) promoter. (B) Representative images depicting cellular localization patterns of RAN-translated GA-GFP aggregates. (C) Western blot analysis of GA-GFP levels in primary neurons using anti-GA antibody (RRID: AB_2728663) (NT = non-transduced; GFP = transduced with a GFP lentiviral vector (negative control); $G_4C_2$-GFP = transduced with RAN translation vector). (D) Schematic of experimental timeline from day in vitro (DIV) 0 to 10. (E) Fluorescent confocal imaging and quantification of DPR aggregate formation in primary neurons transduced with RAN translation vector, then incubated with either normo-glucose media (25 mM glucose + 0 mM 2DG) or media containing increasing concentrations of 2DG ($n = 4$). Mounted coverslips were imaged across >5 randomly selected fields of view/experiments. More than 30 cells were analyzed per condition. (F) qRT-PCR analysis of GFP mRNA levels in primary neurons transduced with either the RAN translation reporter vector, then incubated with either normo-glucose media or 10 mM 2DG-containing media ($n = 4$). (G) Schematic of lentiviral ATG translation vector used for the experiments, containing a 50 GA repeats encoded with alternative codons and a downstream GFP tag. (H) Fluorescent imaging and quantification of DPR aggregate formation in primary neurons transduced with ATG translation vector, then incubated with normo-glucose media or 10 mM 2DG-containing media ($n = 3$). (I) qRT-PCR analysis of GFP mRNA levels in primary neurons transduced with the ATG translation vector, then incubated with normo-glucose media or 10 mM 2DG-containing media ($n = 3$). For (E), one-way ANOVA with Dunnett's test for multiple comparisons. For (F–I), student's two-tailed t-test. All data are presented as mean ± standard error of the mean (SEM; $n = 3$–4 biological replicate). $**p < 0.01$, $****p < 0.0001$, n.s. $p > 0$. Source data are available online for this figure.

## Glucose hypometabolism selectively impairs the survival of i³Neurons from C9orf72 ALS patients through activation of the GCN2 kinase

Given the toxic profile of DPRs, we posit that glucose hypometabolism—by augmenting RAN translation—would cause a higher level of toxicity in neurons carrying the repeat expansion compared to controls. We tested this hypothesis in hiPSC-derived neurons generated via doxycycline-inducible expression of pro-neural transcription factors (Fig. 5A). This approach produces highly consistent and homogenous neuronal cultures (i³Neurons), which express the full complement of neuronal markers and exhibit classic electrophysiological properties of neurons (Fernandopulle et al, 2018). We generated mature i³Neurons from healthy controls and C9orf72 ALS patients, then removed glucose from the cell culture media (Fig. 5B). This approach led to a time-dependent reduction in neuronal activity, which was rescued by the re-addition of glucose (Fig. EV4A). Moreover, as with our 2DG treatment paradigm, we verified that complete glucose deprivation decreases the metabolic rate of the cells (Fig. EV4B), causes ISR activation, and enhances DPR production (Fig. EV4C–F). We then tracked the survival of both control and C9 patient-derived i³Neurons (with three iPS lines per genotype) cultured in glucose-deprived or normo-glucose media over six days and found that C9 neurons had markedly worse survival in glucose-deprived media compared to controls, as well as compared to C9 neurons cultured in normo-glucose (Fig. 5C,D). We measured the GP levels by dot-blot analysis in C9 neurons and determined that they were 2–3 fold higher than the background signal observed in control neurons (absence of C9 expansion) (Figs. 5E and EV5E).

Furthermore, we found that the C9-specific survival defect was ameliorated by culturing neurons with the GCN2 inhibitor A92 (Fig. 5F), which suggests that DPR production *via* GCN2 activation is involved in the C9-specific vulnerability to glucose deprivation. In further support of this, A92 treatment also led to a significant drop in endogenous GP production in glucose-deprived C9 neurons, but not those exposed to normo-glucose (Fig. 5G). The partial rescue in the survival phenotype we observed in the C9 neurons treated with A92 (Fig. 5F), may reflect the involvement of other pathogenic mechanisms caused by the repeat expansion besides DPR toxicity (for example, C9orf72 haploinsufficiency (Wang et al, 2021)).

## Exacerbating energy imbalances increases DPR accumulation and causes motor dysfunction in the C9 BAC transgenic mouse

To corroborate our findings in an in vivo model and explore the potential downstream behavioral and phenotypical consequences of glucose hypometabolism in C9-ALS/ALS, we returned to the C9 BAC transgenic mouse. Despite exhibiting some basal level of dipeptide repeat pathology in several CNS regions, these mice do not manifest ALS-like motor dysfunction (O'Rourke et al, 2015), which we hypothesized is due to *insufficient* production of DPRs. We further hypothesized that exacerbating metabolic dysfunction in these animals would cause significant stress to the CNS, increase DPR production, and cause the onset of a motor phenotype.

To target glucose metabolism in the CNS, we i.p. injected mice with 2DG (Fig. 6a), which O'Connor and colleagues demonstrated to cause ISR activation in the murine CNS (O'Connor et al, 2008). We performed an initial experiment to confirm these results, and indeed, observed a 2DG dose-dependent increase in spinal cord mRNA levels of both CHOP and GADD34 (two downstream targets of the ISR) within 16 h of 2DG injection (Fig. 6B). However, we also observed acute toxicity in several animals treated with the highest dose (8 g/kg), and therefore we proceeded with further experiments using the intermediate dose (4 g/kg). We exposed animals to 4 g/kg 2DG weekly for six weeks. After the treatment course, we again employed LC-MS to confirm the presence of 2DG-6-P (the primary metabolite of 2DG) in the brain and assess the degree of metabolic alteration it caused. As expected, we detected 2DG-6-P accumulation in the brain of 2DG-injected animals (Fig. 6C), as well as an increased concentration of fructose-1,6-bisphosphate and enrichment of metabolites involved pathways such as gluconeogenesis, transfer of acetyl groups into the mitochondria, the citric acid cycle, and several others (Figs. 6D and EV6A; Dataset EV1), which validates this approach as a method to introduce metabolic impairment in the CNS. Interestingly, using confocal scanning laser ophthalmoscopy (cSLO) imaging, we also observed evidence of retinal damage resulting from 2DG treatment. Specifically, compared to saline-treated, the 2DG-treated C9-BAC animals had a higher number of round hyperfluorescent foci (Fig. EV6B), which are commonly associated with various pathological retinal conditions (Fragiotta et al, 2021). This further highlights the CNS effects of the 2DG injection paradigm.

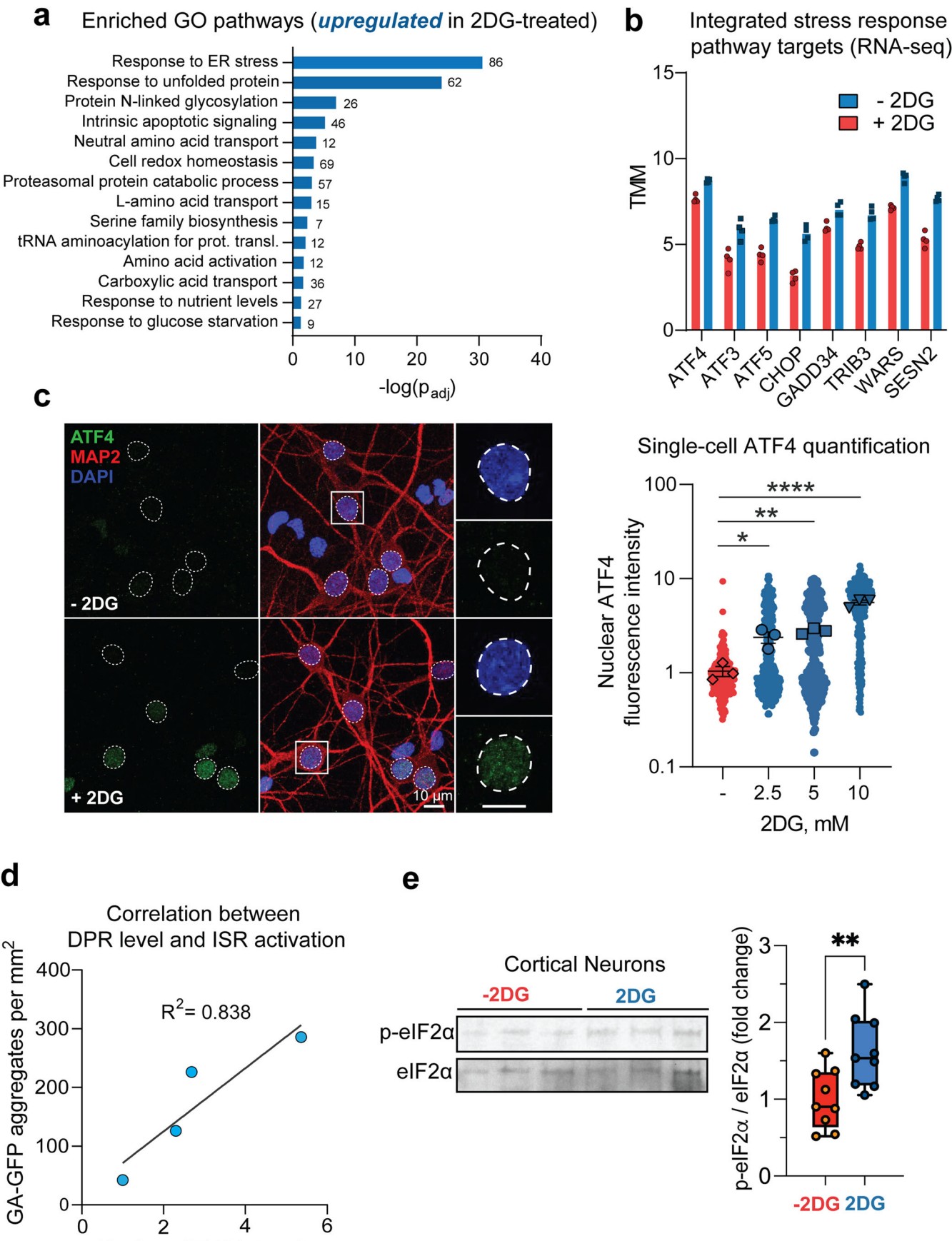

**a** Enriched GO pathways (*upregulated* in 2DG-treated)

**b** Integrated stress response pathway targets (RNA-seq)

Single-cell ATF4 quantification

**c** ATF4 / MAP2 / DAPI

**d** Correlation between DPR level and ISR activation

$R^2 = 0.838$

**e** Cortical Neurons

p-eIF2α
eIF2α

**Figure 3. Glucose hypometabolism activates the ISR in cultured neurons.**

ISR activation is measured in cortical neurons exposed to 2DG. (A, B) RNA sequencing assessment of select transcriptomic changes in C9orf72 patient-derived i³Neurons ($n = 2$ individual i³Neuron lines with 2 separate differentiations per line) incubated in media containing 10 mM 2DG for 48 h versus those maintained in normal media, including significantly upregulated gene ontology (GO) pathways (A) and significantly upregulated individual integrated stress response (ISR) target transcripts (B; all $p_{adj} < 0.05$). (C) Immunofluorescence-based measurement and quantification of nuclear ATF4 expression level in MAP2-positive primary neurons incubated in media containing 0, 2.5, 5.0, or 10 mM 2DG for 48 h ($n = 4$). (D) Correlation between the number of GA-GFP aggregates per field of view (from Fig. 2E) and nuclear ATF4 expression level (from Fig. 3C) in primary neurons incubated with either normo-glucose or various concentrations of 2DG. Best-fit line and $R^2$ value represent linear regression analysis. (E) Western Blot analysis for phospho-eIF2α and eIF2α of cortical neurons treated with glucose or 2DG and relative quantification of the ratio between phospho-eIF2α and eIF2α protein expression ($n = 3$; $m = 3$ technical replicates). For (A), the values adjacent to each bar represent the number of altered genes in each GO pathway. The statistical test applied is the Fischer's exact test with a Benjamini–Hochberg False Discovery Rate (FDR). For (B), bars represent median values, and individual dots represent individual replicates. For (C), one-way ANOVA with Dunnett's test for multiple comparisons. Data are presented as mean ± SEM of biological replicates ($n = 3$). At least 30 neurons were analyzed/condition. *$p < 0.05$, **$p < 0.01$, ****$p < 0.0001$. For box and whisker plots, box edges denote upper and lower quartiles, horizontal lines within each box denote median values, whiskers denote maximum and minimum values, and shaded circles denote individual values for each replicate. Source data are available online for this figure.

But does 2DG-mediated metabolic impairment alter DPR production and/or motor function? We found that chronic 2DG treatment was accompanied by a worsening of motor performance, as indicated by a decreased latency to fall of 2DG-injected C9 animals on the inverted wire hang test (Fig. 6E), which was independent of changes in body weight (Fig. EV6C). Importantly, in wild-type animals, we *did not* detect a deficit in inverted wire hang performance (Fig. EV6D), indicating that the deficit is specific to C9-BAC animals. Furthermore, using single molecule array (Simoa) assay (Fig. 6F), we observed higher GP levels in the spinal cord homogenate of 2DG-injected C9-BAC animals compared to those injected with saline. Higher levels of GP were also detected in 2DG vs. saline-treated C9 mice using a dot blot immunoassay technique (Fig. EV6E). In the spinal cord homogenates of these mice, we also measured by western blot an approximately 60% increase in the phospho/dephospho eIF2α ratio (Fig. EV6F). Together, our data indicate that exacerbating metabolic imbalances and ISR activation in vivo enhances DPR production and causes a motor phenotype in a transgenic mouse model of C9-ALS/FTD.

## PR contributes to neuronal glucose hypometabolism and ISR activation

We also wanted to determine whether RAN-translated DPRs could themselves contribute to neuronal glucose hypometabolism and ISR activation. We again employed synapsin-driven lentiviral vectors to drive DPR production in neurons, but for these experiments, we encoded the DPRs using a randomized codon strategy such that only one of the five possible DPRs would be expressed at any time and using an ATG start codon. We also included a GFP tag downstream of the DPRs to visualize their cellular localization and used a GFP-only vector as a control (Fig. 7A). These vectors drive robust DPR/GFP expression with the expected cellular localization patterns. For example, the GFP fluorescence resulting from the PR vector mainly localizes to the nuclear and nucleolar regions of the cells (Fig. 7B). We transduced cells with equivalent amounts of lentivirus for each construct (which we determined using qRT-PCR-based measurement of lentiviral titers; Fig. EV7A) and found that each vector drove GFP/DPR expression with similar transduction efficiencies ranging from 85% to 95% (Fig. EV7B,C).

We then measured how these DPRs affect the neurons' ability to take up and metabolize glucose. Interestingly, using a luminescence-based glucose uptake assay (which we first validated for use with neurons; Fig. EV7D), we found that expression of PR

(but not the other DPRs) caused a ~25% reduction in glucose uptake normalized to total protein, compared to the GFP control (Fig. 7C). Furthermore, using the Seahorse flux assay, we found that expression of PR *also* caused a significant reduction in extracellular acidification rate (ECAR) normalized to total protein (Fig. 7D), which is an indicator of the rate of glycolysis occurring in the cells. Therefore, our data indicate that the presence of PR in neurons is sufficient to impair glucose metabolism. We also detected a trend towards decreased oxygen consumption in PR-expressing neurons, indicating that impaired glucose metabolism caused by PR decreases the amount of pyruvate available for oxidative phosphorylation (Fig. EV7E). As expected, we also found that PR-expressing neurons had a significantly higher nuclear ATF4 level (Fig. 7E), which indicates that PR can *also* activate the ISR, potentially through its effect on glucose metabolism, although we cannot rule out the other mechanisms initiated by PR. Still, these data indicate that PR can independently contribute to neurons' energy imbalance and ISR activation in neurons.

## Discussion

In this study, we identified a novel link between energy imbalance and disease pathogenesis in C9orf72-ALS/FTD, potentially revealing some therapeutic intervention opportunities. Specifically, we demonstrated that brain glucose hypometabolism—which is a consistently observed phenomenon that occurs years before symptom onset in C9orf72 patients but also in other neurodegenerative diseases (Cunnane et al, 2020; De Vocht et al, 2020; Popuri et al, 2021)—can act as a modifier of RAN translation and ultimately prime and exacerbate disease phenotypes, both in in vitro and in vivo models of disease. These findings are particularly important because they indicate that, by heightening the production of pathogenic DPRs, pre-symptomatic metabolic imbalances may play a central role in initiating or potentiating disease. This, in turn, highlights a critical window for therapeutic intervention to mitigate RAN translation and, thereby, possibly delaying disease onset or slowing disease progression. Many FDA-approved metabolic treatments already exist (Kinch et al, 2015) and could quite easily be tested in pre-clinical models of C9orf72 ALS/FTD and eventually in patients.

Our research underscores the critical need for further pre-clinical investigations to elucidate the consequences of maintaining bioenergetic homeostasis in C9orf72 ALS/FTD model systems, particularly in symptomatic and pre-symptomatic stages. The

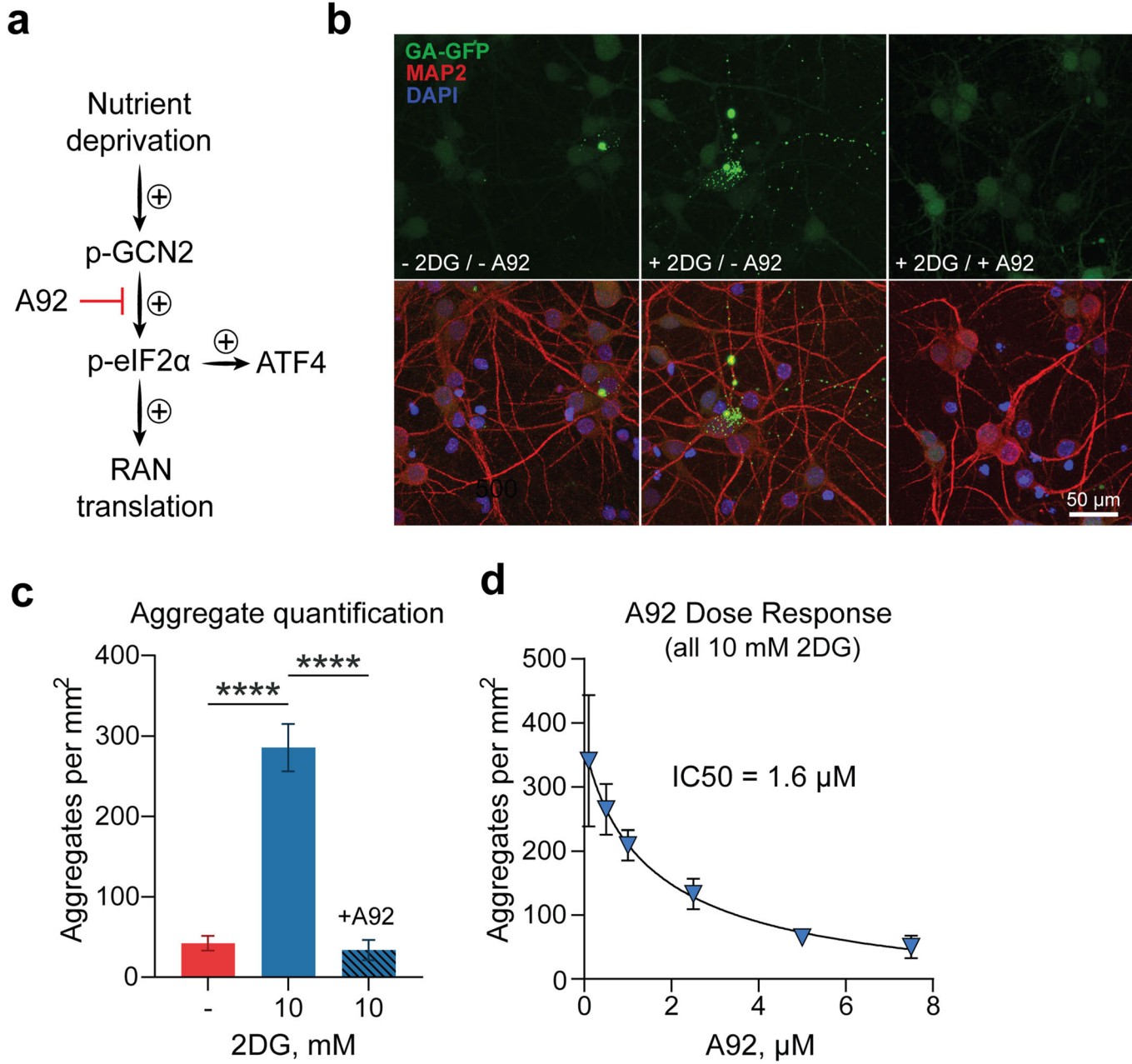

**Figure 4. DPR accumulation caused by glucose hypometabolism correlates with ISR activation and is blocked by inhibition of the GCN2 kinase.**

DPRs aggregation in cortical neurons is measured in hypometabolic conditions. (A) Schematic depicting the proposed mechanism through which nutrient deprivation increases RAN translation. A92 was used as a pharmacological inhibitor of GCN2 kinase activity. (B, C) Fluorescent confocal imaging and quantification of DPR formation in primary neurons transduced with RAN translation vector, then incubated with either normo-glucose media, 10 mM 2DG-containing media, or 10 mM 2DG-containing media with 5.0 μM A92, all in the presence of 0.1% DMSO ($n = 4$). (D) Quantification of DPR aggregates formation in primary neurons transduced with RAN translation vector, then treated with A92 concentrations ranging from 0.1 to 7.5 μM, all in the presence of 10 mM 2DG and 0.1% DMSO ($n = 4$). Best-fit line and $IC_{50}$ value were derived from non-linear regression analysis. For (C), one-way ANOVA with Dunnett's test for multiple comparisons. Images were taken across 5 randomly selected fields. All data are presented as mean ± SEM ($n = 4$ biological replicates). ****$p < 0.0001$. Source data are available online for this figure.

rationale for exploring both pre-symptomatic and symptomatic conditions lies in the potential to unveil temporal nuances in the interplay between bioenergetic homeostasis and disease pathogenesis. Investigating the pre-symptomatic phase offers a unique opportunity to identify early molecular events and subtle changes that may precede the onset of clinical symptoms. This knowledge

could be instrumental in developing interventions that target disease mechanisms at their incipient stages, potentially altering the course of C9orf72 ALS/FTD progression.

Our findings also highlight another potential therapeutic target for C9orf72-ALS/FTD: the integrated stress response (ISR). In support of this, we observed activation of the ISR in neurons

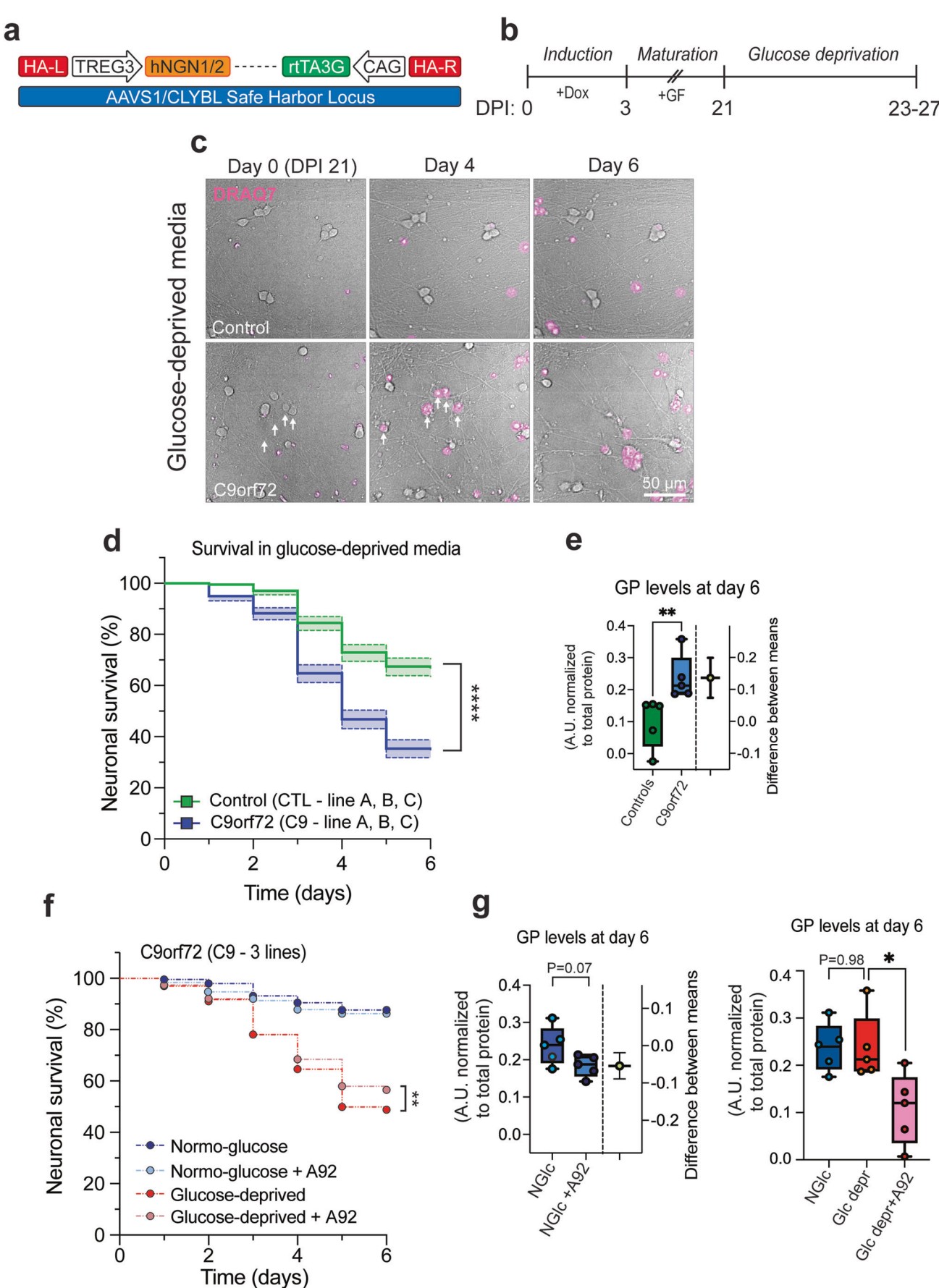

**Figure 5. Glucose deprivation is selectively toxic to C9orf72 patient-derived i³Neuron.**

i³Neurons from patient and control cases are exposed to glucose deprivation and neuronal viability over time is recorded. (A) Schematic of the doxycycline-inducible neuronal differentiation cassette used to drive rapid differentiation of human induced pluripotent stem cells (hiPSCs) into i³Neurons from days post-induction (DPI) 0 to 27. (B) Timeline of differentiation into i³Neurons and subsequent glucose deprivation. (C) Live-cell longitudinal imaging of healthy control- or C9orf72 patient-derived i³Neurons cultured with glucose-deprived media over 6 days using DRAQ7 as a dead cell indicator. (D) Kaplan–Meier survival analysis of i³Neurons derived from either C9orf72 patients or healthy controls and maintained in glucose-deprived media ($n = 3$ i³Neuron lines per genotype with 3 independent differentiations per line). (E) Quantification of dot blot for GP performed on i³Neurons derived from either C9orf72 patients or healthy controls and maintained in glucose-deprived media. (F) Kaplan–Meier survival analysis of C9orf72 patient-derived i³Neurons maintained in either normo-glucose or glucose-deprived media and treated with either 2.5 µM A92 with 0.1% DMSO or 0.1% DMSO only as vehicle control ($n = 3$ i³Neuron lines per treatment with 3 independent differentiations per line). (G) Quantification of dot blot for polyGP performed on C9orf72 patient-derived i³Neurons maintained in either normo-glucose or glucose-deprived media and treated with either 2.5 µM A92 with 0.1% DMSO or 0.1% DMSO only as vehicle control ($n = 3$ i³Neuron lines per treatment with 3 independent differentiations per line). Kaplan–Meier log-rank survival test. *$p < 0.05$, **$p < 0.01$, ****$p < 0.0001$. Neurons were imaged live across >5 randomly selected fields of view/conditions. The same field of view was imaged daily. More than 100 neurons were tracked/condition/experiment. All data are presented as mean ± 95% CI ($n = 3$ biological replicates). For box and whisker plots, box edges denote upper and lower quartiles, horizontal lines within each box denote median values, whiskers denote maximum and minimum values, and shaded circles denote individual values for each replicate. Source data are available online for this figure.

exposed to energy deprivation and in the spinal cord of C9orf72-ALS/FTD patients, highlighting the clinical relevance of ISR activation. But more importantly, we also found that pharmacological inhibition of GCN2—which is one of the four kinases that controls ISR activation (Pakos-Zebrucka et al, 2016)—completely blocked DPR accumulation caused by glucose hypometabolism, which suggests that GCN2 inhibition may serve as a viable therapeutic strategy to attenuate RAN translation caused by metabolic imbalances. Of note, ALS patients are subject to numerous other forms of stress in addition to that caused by energy imbalance (Masrori and Van Damme, 2020), and therefore, the potential contributions of the other three stress-sensing ISR kinases (i.e., PERK, PKR, and HRI) should also be considered. Indeed, several studies have identified roles for PERK and PKR in mediating RAN translation caused by oxidative stress and/or (GGGGCC) repeat-containing RNA, respectively (Cheng et al, 2018; Green et al, 2017; Westergard et al, 2019; Zu et al, 2020). Moreover, perturbation to the mitochondria (also commonly seen in ALS) was recently found to activate HRI, the fourth ISR kinase (Guo et al, 2020). Therefore, rather than selective inhibition of individual ISR kinases, a comprehensive approach focusing on the direct reversal of eIF2α phosphorylation may be more appropriate. In experiments in vitro, we observed a partial rescue of RAN translation with the ISR inhibitor ISRIB. However, this rescue was incomplete, suggesting that other pathways known to respond to energy imbalance (for example, the mTOR pathway (Leprivier and Rotblat, 2020)) may also be involved. Additional studies are warranted to understand the full extent of the role of the ISR and other interconnected pathways as mediators of RAN translation and determine the most efficacious strategies to mitigate DPR production caused by energy imbalance and other forms of cellular stress.

We also identified significant brain energetic and metabolic alterations at baseline—in the absence of pharmacological manipulation—in the frontal cortex of 6-month-old C9-BAC mice. Specifically, we observed reduced ATP levels (and ATP:ADP/ATP:AMP ratios) and alterations in glucose metabolic pathways such as glycolysis and gluconeogenesis. We also observed increased amino acid metabolism in C9-BAC animals, which may reflect a compensatory shift towards using amino acids as alternative substrates for oxidative metabolism. In line with these results, a recent study identified an increase in brain microvascular expression of GLUT1—the sole glucose transporter present at the blood-brain barrier—also in an asymptomatic C9-BAC mouse

model (Pan et al, 2022), which may reflect an attempt to compensate for altered brain glucose utilization. Based on these lines of evidence, the repeat expansion is sufficient to perturb brain energy homeostasis in vivo. The fact that this occurs in the absence of motor and cognitive behavioral phenotypes and overt neuropathology is somewhat surprising—however, these data mirror clinical findings showing that glucose hypometabolism can manifest years before disease onset (De Vocht et al, 2020; Popuri et al, 2021). C9orf72-related metabolic imbalances may accumulate without consequence for some time—potentially years in humans—before becoming disease-causative. Longitudinal metabolomic and transcriptomic studies would be useful to fully characterize the extent of metabolic imbalances caused by the $G_4C_2$ repeat expansion and may provide additional insight into how they accumulate or evolve. Potential non-cell-autonomous contributions to C9orf72-related energy imbalances would also be worth investigating, especially since C9orf72 patient-derived astrocytes exhibit metabolic inflexibility and may have an impaired capacity to provide metabolic support to neurons (Allen et al, 2019). We reported worsening motor phenotypes in C9-BAC mice treated with 2DG. However, the interpretation of these results remains complicated by the pre-existing metabolic imbalances we found in the CNS of these mice. This raises valid concerns about the logical inference from combining two detrimental factors, the disease-causative genetic mutation and the pharmacologically induced glucose hypometabolism. While our study has some intrinsic limitations, we recognize that the reciprocal relationship between RAN translation and GCN2-dependent stress cascades in ALS pathogenesis may require further exploration, particularly using in vivo experimental approaches. To further reinforce the strength and reliability of our study, we should be considering the evaluation of disease-relevant phenotypes in mice through GCN2 ablation or drug interventions in future investigations.

Another question our study raised is how the repeat expansion triggers metabolic imbalance in vivo. Since the C9-BAC mouse is a gain-of-function model and lacks neurodegeneration (which could confound metabolite measurements), the perturbations we identified in this study must be due to the gain-of-function of the $G_4C_2$ repeat expansion. A primary gain-of-function mechanism is DPR production through RAN translation, and in fact, we also found that the arginine-rich DPRs can hinder glucose metabolism in cultured neurons. Furthermore, several other studies have found that GR localizes to the mitochondria and compromises mitochondrial function (Choi et al, 2019; Li et al, 2020; Lopez-Gonzalez et al,

**a**

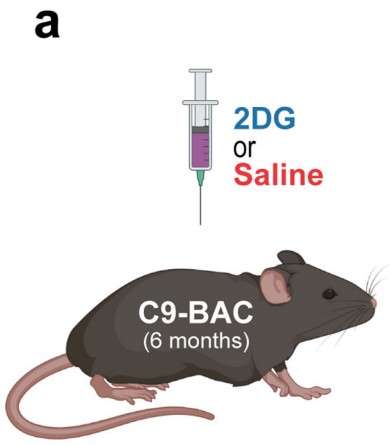

**b**

### CHOP (RT-qPCR)

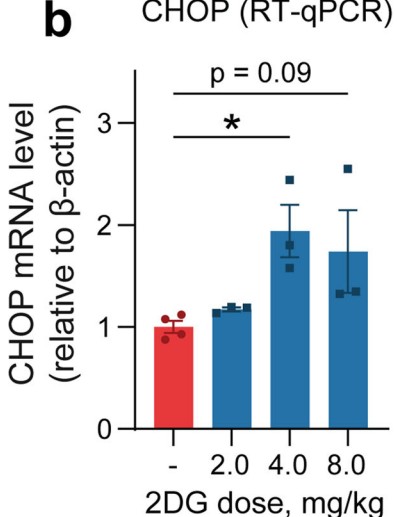

### GADD34 (RT-qPCR)

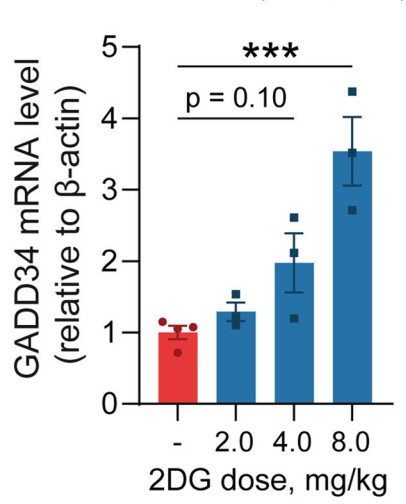

**c**

### 2DG-6-P brain accumulation

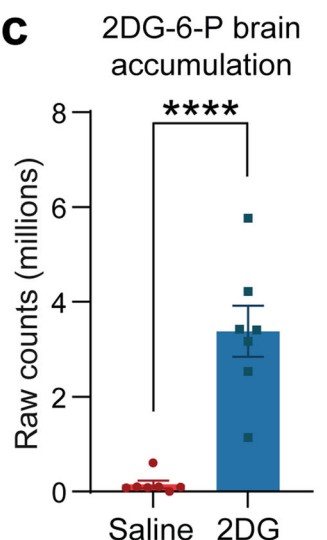

**d**

### Altered metabolites in the brain of 2DG-injected vs. saline-injected (C9-BAC animals)

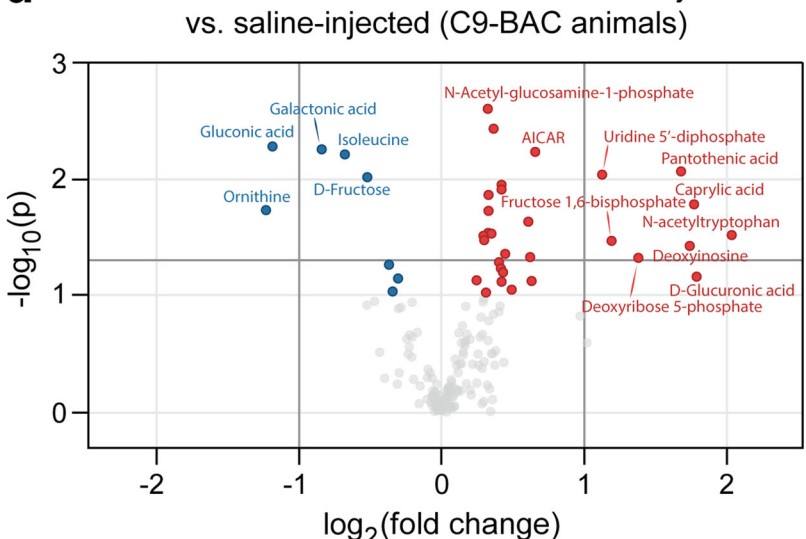

**e**

### Inverted wire hang test

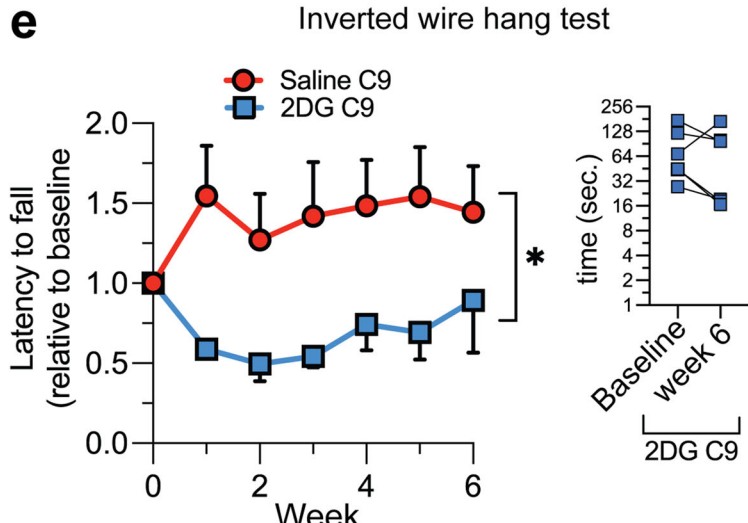

**f**

### GP Simoa Assay (spinal cord lysates)

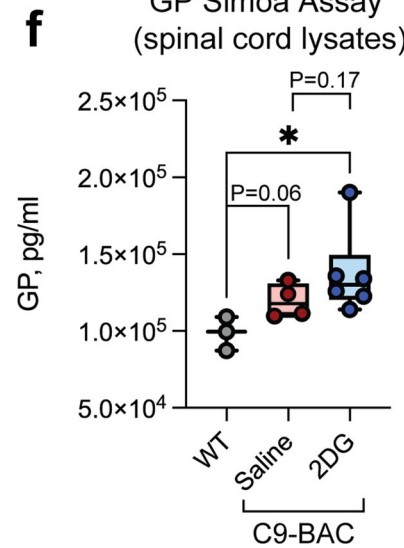

◄ **Figure 6.   2DG treatment exacerbates metabolic stress and drives disease-related phenotypes in C9orf72 BAC transgenic mice.**

Effects of 2DG treatment on metabolic stress and disease phenotypes in C9orf72 transgenic mice. (A) Schematic of experimental setup: C9-BAC animals were treated with 2-deoxyglucose (2DG) or saline (vehicle control) by i.p. injection. (B) RT-qPCR measurement of spinal cord mRNA levels of two ISR transcriptional targets (CHOP and GADD34) in C9-BAC animals acutely treated with various doses of 2DG or saline ($n = 3$–4 animals per condition). (C, D) LC-MS measurement of 2DG-6-P levels (C) and significantly altered metabolites (D) in the frontal cortex of C9-BAC mice following chronic weekly exposure to 4 g/kg 2DG versus saline ($n = 7$ animals per condition). Dark shaded lines indicate significance cut-offs ($-\log_{10}(p) > 1.3$ and $\log_2(FC) > |1|$). (E) Longitudinal assessment of inverted wire hang performance of C9-BAC animals chronically exposed to 4 g/kg 2DG or saline ($n \geq 5$ animals/condition). The inset figure displays a longitudinal assessment of the 2DG treatment effects on the C9-BAC mice. Baseline measurements represent the initial state, followed by longitudinal measurements at 6 weeks post-treatment, illustrating the impact of 2DG intervention on C9-BAC mice over time. Y-axis in Log 2 scale. (F). Single molecule array (Simoa) measurement of 8 M urea-soluble GP levels relative to total protein levels in the spinal cord of saline-treated wild type animals ($n = 3$), saline-treated C9-BAC animals ($n = 4$), or 2DG-treated C9-BAC animals ($n = 6$). For (B), one-way ANOVA with Dunnett's test for multiple comparisons. For (C, D), student's two-tailed t-test. For (E), two-tailed t-test with Welch's correction. For (F), one-way ANOVA. All data are presented as mean ± SEM ($n > 3$ biological replicates). *$p < 0.05$, ***$p < 0.001$, ****$p < 0.0001$. For box and whisker plots, box edges denote upper and lower quartiles, horizontal lines within each box denote median values, whiskers denote maximum and minimum values, and shaded circles denote individual values for each animal. Source data are available online for this figure.

2016). Therefore, we speculate that DPR production may contribute to the metabolic deficits we identified in vivo, although further studies will be required to confirm this. It is also worth noting that, in humans, in addition to causing RAN translation, the repeat expansion also causes haploinsufficiency of the endogenous C9orf72 protein, which is potentially relevant to energy metabolism because the loss of C9orf72 was recently found to de-stabilize the electron transport chain (ETC) and in turn disrupt mitochondrial ATP production (Wang et al, 2021). Therefore, future studies should aim to elucidate the potential synergistic roles of *both* gain-of-function mechanisms (i.e., DPR-mediated disruption of glycolysis) and loss-of-function mechanisms (i.e., de-stabilization of the ETC), as both would be expected to occur in human C9orf72-ALS/FTD carriers and contribute to energy imbalance.

In summary, our data point towards a pathogenic feedforward loop in which stress from ALS-associated energy imbalances enhances RAN translation and accumulation of DPRs, exacerbating glucose hypometabolism and metabolic stress. In essence, brain energy metabolic decline, a feature of C9orf72-ALS/FTD but also observed in other neurodegenerative conditions, drives and is driven by neurodegeneration and neuronal impairment in a potentially destructive cycle. Therapeutically targeting this neurotoxic feedforward loop—either by correcting the energy imbalance or mitigating stress response activation—may be a powerful approach to mitigate neurodegeneration in C9orf72-ALS/FTD.

## Methods

### Targeted polar metabolite profiling

Metabolites from equal amounts of frontal cortex tissue were rapidly extracted in 80% ice-cold methanol. Extracted samples were vortexed twice, cleared by centrifugation at $14,000 \times g$ for 20 min at 4 °C, and stored at −80 °C. The Weill Cornell Medicine Meyer Cancer Center Proteomics & Metabolomics Core Facility performed hydrophilic interaction liquid chromatography-mass spectrometry (LC-MS) for relative quantification of targeted polar metabolite profiles. Metabolites were measured on a Q Exactive Orbitrap mass spectrometer, coupled to a Vanquish UPLC system by an Ion Max ion source with a HESI II probe (Thermo Scientific). A Sequant ZIC-pHILIC column (2.1 mm i.d. × 150 mm, particle size of 5 μm, Millipore Sigma) was used for separation. The MS data were processed using XCalibur 4.1 (Thermo Scientific) to obtain the metabolite signal intensity for relative quantitation. Targeted

identification was available for 205 metabolites based on an in-house library established using known chemical standards. Identification required exact mass (within 5 ppm) and standard retention times. Relative metabolite abundance data were median-normalized and log-transformed, and differential abundance and pathway analyses were done with the free online tool MetaboAnalyst 5.0 (Pang et al, 2021). Metabolite significance was determined with one-way ANOVA with posthoc t-tests, with the cutoff being a raw $p$-value < 0.05, and the pathway significance cutoff was an FDR < 0.05.

### Lentiviral transfer vector construction and lentivirus preparation

For the RAN translation transfer vector, the 5' region of the *C9orf72* gene and the GFP tag were PCR-amplified from previously described plasmids (Wen et al, 2014; Westergard et al, 2019), then inserted into an AgeI-/EcoRI-linearized pLenti-hSyn backbone (Addgene #86641) by Gibson assembly. The (GGGGCC)$_{188}$ repeat expansion was generated using previously described protocols (Wen et al, 2014), then introduced by classical restriction cloning. For each codon-optimized DPR$_{50}$ transfer vector, the DPR-GFP sequences were PCR-amplified from previously described plasmids (Wen et al, 2014), then inserted into an AgeI-/EcoRI-linearized pLenti-hSyn backbone (Addgene #86641) by Gibson assembly. The complete cloning strategies, primers (GeneWiz), and restriction enzymes (NEB) used are contained within the corresponding .dna files. All PCR products were amplified using a standard Q5 polymerase protocol (NEB) and purified with an agarose gel extraction kit (Thermo Scientific). Gibson assemblies and ligations were performed per the manufacturer's recommendations (Thermo Scientific). The sequences of all plasmids were verified by Sanger sequencing (GeneWiz).

Lentivirus was prepared by co-transfection of a confluent 15 cm plate of HEK 293-FT cells (Invitrogen) with 8 μg of transfer vector, 16 μg of packaging plasmid (psPAX2; Addgene #12260), and 4 μg of envelope plasmid (pMD2.G; Addgene #12259) with 112 μg PEI MAX (Polysciences, Inc.) as a transfection reagent. Forty-eight hours later, the viral supernatant was harvested and filtered through a 0.45 μm PVDF syringe filter. Lentivirus was then concentrated using Lenti-X Concentrator (Takara Bio), reconstituted in PBS, and the resulting titers estimated using the Lenti-X qRT-PCR Titration Kit (Takara Bio), all following manufacturers' recommendations. Concentrated lentivirus was aliquoted and stored at −80 °C.

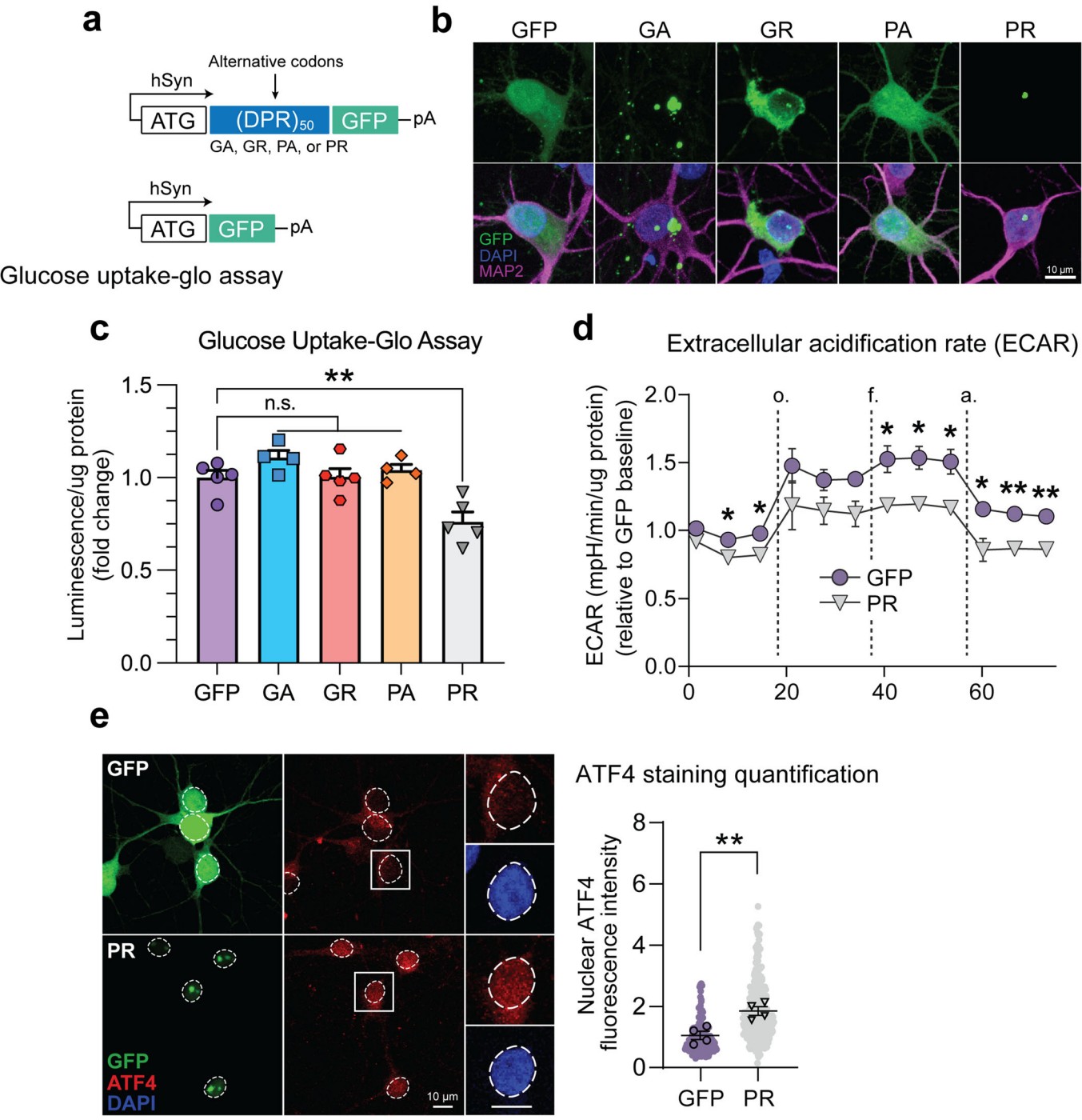

**Figure 7. Arginine-rich DPRs contribute to glucose hypometabolic stress.**

Glucose uptake is measured in cortical neurons expressing DPRs. (A) Schematic of lentiviral DPR vectors and GFP-only (control) vector. (B) Representative images depicting cellular localization patterns of GFP-tagged DPRs in rodent primary neurons transduced with lentiviral vectors. (C) Luminescence-based measurement of glucose uptake (normalized to total protein and expressed in fold change) in primary neurons transduced with each of the DPR vectors or the GFP-only control vector ($n = 4$ biological replicates). (D) Seahorse extracellular flux assay measurement of extracellular acidification rate (ECAR; normalized to total protein) of primary neurons transduced with either the PR vector or GFP-only control vector ($n = 3$ biological replicates). (E) Immunofluorescence-based measurement and quantification of nuclear ATF4 expression level in MAP2-positive primary neurons transduced with either the PR vector or GFP-only control vector ($n = 4$ biological replicates). At least 30 neurons were analyzed/condition. For (C), one-way ANOVA with Dunnett's test for multiple comparisons. For (D), multiple student's two-tailed t-tests. For (E), student's two-tailed t-test. All data are presented as mean ± SEM. $*p < 0.05$, $**p < 0.01$. Source data are available online for this figure.

## Primary neuron preparation

Embryonic day 16 (E16) rat embryos were harvested, brains were dissected, and meninges removed. Cortical and midbrain regions were cut into small pieces and digested by exposure to 0.2% trypsin for 45 min at 37 °C with gentle shaking, after which fetal bovine serum (FBS) was added to neutralize the digestion. The tissue was washed several times with HBSS and passed through a 70 μm strainer to create a single-cell suspension. The neurons were then plated in a neurobasal medium supplemented with B-27, penicillin-streptomycin, and L-glutamine on plates pre-coated with poly-D-lysine or a combination poly-D-lysine and laminin (for glass coverslips). Half media changes were performed twice weekly.

## Induced pluripotent stem cell (iPSC) maintenance and differentiation

iPSC lines were obtained as generous gifts from the labs of Drs. Ward and Barmada. We received these lines already genetically edited to contain doxycycline-inducible neuronal differentiation cassettes in the AAVS1 and CLYBL safe harbor loci (Fig. 5A). Standard human iPSC maintenance and differentiation protocols were adapted from a previous study (Fernandopulle et al, 2018) and are described below. Information about iPSC lines are provided in Dataset EV2.

### General maintenance

The human iPSCs were grown as colonies on Matrigel-coated (Corning) 10 cm plates in Essential 8 (E8) medium (Fisher Scientific), which was changed daily. Spontaneous differentiation was identified under an Evos microscope and removed twice weekly by mechanical dissociation. Once reaching a confluency of ~80%, the colonies were removed from the plate by gentle scraping, then split 1:2 into new plates, with care taken to ensure the colonies remained intact. 10 μM rock inhibitor (Selleck Chemicals) was added to the media after splitting.

### Differentiation into i³Neurons

Differentiation was accomplished in two phases: (1) an induction phase from days post-induction (DPI) 0–3, in which doxycycline (Sigma) was added to the media to facilitate gene induction and (2) a maturation phase (from DPI 3 and onwards), in which the cells were exposed to neurotrophic factors and allowed to differentiate to maturity. To initiate induction, hiPSCs were dissociated into a single-cell suspension with accutase (Fisher Scientific), re-plated onto matrigel-coated 10 cm plates (1–2 million cells per plate), and switched to doxycycline-containing "induction medium" consisting of DMEM/F12 with N2 supplement, glutamax, and NEAA (all Fisher Scientific). Induction media was replaced daily. For the maturation phase, the induced cells were re-plated onto poly-L-ornithine- (Sigma) and laminin- (Fisher Scientific) coated plates and switched to a "maturation medium" consisting of BrainPhys Neuronal Medium (StemCell Technologies) with B-27 supplement (Fisher Scientific), BDNF/NT3 (PeproTech), and laminin (Fisher Scientific). Half media changes with maturation media were performed twice weekly until the cells were utilized for experiments at DPI 21-27.

## Neuronal transductions and treatments

At day in vitro (DIV) 8 (for primary neurons) and DPI 21 (for i³Neurons), lentiviral particles were diluted into fresh neuronal media at a concentration of $2 \times 10^7$ lentiviral RNA copies/ml. The cells' media was then replaced with this virus-containing media—72 h later, the virus-containing media was replaced with fresh media.

For 2DG treatments, the cells were cultured with media containing 2DG (Cayman Chemical). Notably, the concentration of glucose (usually 25 mM) was adjusted accordingly to prevent osmotic stress (e.g., if media was prepared with 10 mM 2DG, the concentration of glucose was reduced to 15 mM), which was achieved by adding the desired concentrations of glucose/2DG to glucose-free Neurobasal (Fisher Scientific), then preparing the media as usual. Similarly, glucose-deprived media was prepared by replacing the glucose in the media with 25 mM mannitol (a metabolically inert compound; Sigma).

## Immunofluorescence and confocal imaging analysis

Neurons grown on glass coverslips were washed once with PBS, then fixed with 4% PFA for 15 min. The cells were washed with PBS twice more, then permeabilized with 0.1% Triton-X for 15 min, blocked with 5% bovine serum albumin (BSA) for 45 min, then incubated overnight in a primary antibody (anti-ATF4, Cell Signaling Technology Cat# 11815, RRID:AB_2616025, 1:500; anti-MAP2, Novus Biologicals NB200-213 RRID: AB_2138178, 1:5000) diluted in 1% BSA. The following day, the cells were washed twice with PBS, then incubated for 45 min in the appropriate fluorophore-conjugated secondary antibody (Alexa Fluor Goat anti-Rabbit 546, Thermo Fisher Scientific Cat# A-11035, RRID:AB_2534093, 1:1000; Alexa Fluor Goat anti-Chicken 647 Thermo Fisher Scientific Cat# A-21449, RRID:AB_2535866, 1:1000; Alexa Fluor Goat anti-Rabbit 488 Thermo Fisher Scientific Cat# A-11008, RRID:AB_143165, 1:1000; Alexa Fluor Goat anti-Chicken 488 Thermo Fisher Scientific Cat# A-11039, RRID:AB_2534096, 1:1000; Alexa Fluor Goat anti-Chicken 546 Thermo Fisher Scientific Cat# A-11040, RRID:AB_2534097, 1:1000) in 1% BSA. After two additional washes with PBS and one wash with ultra-pure water, coverslips were mounted onto glass slides using Vectashield with DAPI (Vector Laboratories, H-1200-10) and sealed with clear nail polish.

Mounted coverslips were imaged in the appropriate fluorescent channels using Nikon A1R confocal microscope. Z-stack images (0.3–1.0 μm step size) were taken across >5 randomly selected fields of view per experiment with either 20× or 60× objectives.

Maximum intensity projections were generated and analyzed using ImageJ software. DPR puncta were quantified using the ImageJ multi-point tool. ATF4 nuclear expression was measured using ImageJ by first creating regions of interest (ROIs) based on DAPI staining (for MAP2+ cells only) and then measuring ATF4 mean fluorescence intensity within each ROI. >30 randomly selected cells were analyzed per condition. GFP mean fluorescence intensity was measured similarly, but instead using MAP2 staining to define ROIs.

## Time-lapse imaging and analysis

i³Neurons were differentiated to DPI 21 on glass-bottom plates, then switched to either normo-glucose or glucose-deprived media

containing the viability dye DRAQ7 (Abcam ab109202, 1:100). Immediately following the media change, the cells were imaged live using the Nikon Eclipse Ti microscope in both the brightfield and Cy5 channels across >5 randomly selected fields of view per condition. The same fields of view were imaged daily for the remainder of the experiment. Individual cells were tracked over time and marked dead when they accumulated DRAQ7 fluorescence. >100 cells were tracked per condition per experiment.

## RNA analyses

Total RNA was extracted from either cultured neurons ($>1.5 \times 10^6$) or mouse spinal cord tissue (10–20 mg) using the PureLink RNA Mini kit (Thermo Scientific) and treated with DNase I (NEB) according to the manufacturers' recommendations. RNA concentration and purity (OD 260/280 ratio) were assessed by Nanodrop.

### Real-time PCR
300 ng of total RNA was reverse transcribed using SuperScript IV Reverse Transcriptase (Thermo Scientific). The resulting cDNA was then used for real-time PCR (15 ng per reaction) with the following TaqMan assays: human CHOP, human GADD34, human 18S rRNA, mouse CHOP, mouse GADD34, mouse β-actin, rat β-actin, and GFP (all Thermo Scientific). The resulting $C_t$ values were analyzed using the $2^{\Delta\Delta Ct}$ method and expressed as fold change.

### RNA sequencing
Total RNA from two independent differentiations of two C9orf72 patient-derived i³Neuron lines ($n = 4$) per condition were used. Samples with a minimum RNA integrity number (RIN) of 8.0 were used for library preparation, transcriptome sequencing, differential expression analysis, and gene ontology (GO) enrichment analysis, all of which were performed by Novogene Co., LTD (Beijing, China). An adjusted $p$-value $< 0.05$ was set as the threshold for significance.

## Glucose uptake assays

Primary rat neurons were plated in standard plastic 96-well plates at a density of 40,000 cells/well, then transduced with codon-optimized lentiviruses as described in the previous section. Four days after transduction, the Glucose Uptake-Glo assay (Promega) was performed according to the manufacturer's instructions. The reactions were then transferred to opaque white 96-well plates, and the luminescence was measured using standard luminometer settings on the Cytation 5 plate reader (BioTek). An aliquot of Glucose Uptake Glo assay lysates was analyzed for total protein content using the Pierce 660 nm protein assay according to the manufacturer's instructions and with Ionic Detergent Compatibility Reagent (Thermo Scientific). Luminescence values were then normalized to total protein.

## Seahorse extracellular flux assays

Primary rat neurons or human i³Neurons were plated on Seahorse XFp plates (Agilent) at a density of 40,000 cells/well, then either treated with 2DG or transduced with codon-optimized lentiviruses as described in previous sections. 48 h after 2DG treatment or 4 days after transduction, the Seahorse Extracellular Flux assay was performed using the Seahorse XF Mini instrument (Agilent)

according to the manufacturer's instructions. During the assay, the cells were treated sequentially with each of the following mitochondrial toxins: oligomycin (1.5 μg/ml), FCCP (3 μM), and antimycin (1 μM) (all Cayman Chemical). Immediately after the assay, the cells were lysed with RIPA buffer with protease inhibitor (Thermo Scientific). The lysates were analyzed for total protein content using the Pierce BCA protein assay (Thermo Scientific). Extracellular acidification rate (ECAR) and oxygen consumption rate (OCR) values were then normalized to total protein.

## Lysate preparation for immunoassays

### Human tissues
100 mg of fresh-frozen spinal cord tissue was submerged in 1% SDS buffer with protease inhibitor (Fisher Scientific) and homogenized with a Dounce homogenizer. The resulting lysates were centrifuged at 3000 rcf for 20 min at 4 °C.

### Mouse tissues
15–25 mg of fresh-frozen spinal cord tissue was submerged in 1% SDS buffer with protease inhibitor, triturated with a pipette, and homogenized with a handheld homogenizer. Lysates were then sonicated at 12 °C with a tip sonicator 3–4x for 10 s each (until clear) and centrifuged at 18,000 rcf for 5 min at room temperature. *For GP immunoassays:* samples were processed following the protocol above using 8 M urea/Triton X100 instead of 1% SDS.

### Cultured cells
1–2 million cells in 6-well culture plates were washed with ice-cold PBS, submerged with 150 μl RIPA buffer with protease inhibitor, and placed on an orbital shaker at 4 °C for 15 min. Lysates were transferred to fresh tubes, sonicated with the Bioruptor Pico-Diagenode, and centrifuged at 18,000 rcf for 5 min at 4 °C.

### GP immunoassays
2–3 μg of protein were diluted in 200 μL of 8 M urea and then used for dot blot.

For all lysates, the clarified supernatant was moved to fresh tubes, and Pierce BCA protein assay (Thermo Scientific) was used to measure the protein content of each sample. If used for Western blotting, the samples were mixed 1:1 with 2X Laemmli sample buffer (Bio-Rad) with 5% β-mercaptoethanol (Sigma) and heated to 95 °C for 5 min. Samples (with or without sample buffer) were then aliquoted and stored at −80 °C.

## Immunoblotting

For Western blots, 10–30 μg of protein extracts were run at 100 V for 1 h on 10% pre-cast SDS-PAGE gels (Bio-Rad), then electro-transferred to 0.2 μm nitrocellulose membranes using the Trans-Blot Turbo Transfer system (Bio-Rad). For dot bots, 2–3 μg of protein extracts were blotted directly onto 0.2 μm nitrocellulose membranes and allowed to air dry for 30 min. Membranes were then blocked for 1 h with 5% milk, incubated with primary antibody (anti-C9orf72/C9RANT (poly-GA), Millipore Cat# MABN889, RRID:AB_2728663, 1:500; anti-GFP, Proteintech Cat# 50430-2-AP, RRID:AB_11042881, 1:2000; anti-ATF4, Cell Signaling Technology Cat# 11815 (also ENCAB306IYD),

RRID:AB_2616025, 1:1000; anti-Histone H3, Cell Signaling Technology Cat# 4499, RRID:AB_10544537; Cell Signaling Technology Cat# 2103, RRID:AB_836874 1:500; anti-Phospho-eIF2α (Ser51) Cell Signaling Technology Cat# 9721, RRID:AB_330951 1:500) diluted in 5% BSA (either overnight at 4 °C or 30 min at room temperature), washed 3× with TBS-T, incubated with HRP-conjugated secondary antibody (Donkey anti-Rabbit HRP, Cytiva Cat# NA9340V RRID: AB_772191, 1:10,000; Donkey anti-Mouse HRP, Cytiva Cat# NA9310V RRID: AB_3095979, 1:10,000) diluted in 5% milk for 45 min at room temperature, and washed again 3× with TBS-T. Membranes were developed with SuperSignal West Femto Maximum Sensitivity substrate (Thermo Scientific). Images were acquired using the ChemiDoc XRS+ system (Bio-Rad).

### Single-molecule array (Simoa) immunoassay

A custom "Homebrew" Simoa GP assay was performed using the SR-X Simoa platform (Quanterix). Mouse spinal cord lysates (in 8 M urea buffer) were adjusted to 1 mg/ml protein and 4 M urea; serial dilutions of recombinant $GP_8$ (custom synthesized by Vivitide) were also prepared in 4 M urea buffer. All samples were then diluted 1:10 with Simoa Lysate Diluent C (Quanterix). Then, on a 96-well Simoa assay plate, samples were mixed with: (1) anti-GP monoclonal antibody (Developmental Hybridoma Studies Bank; TALS 828.179)-coated paramagnetic capture beads, and (2) biotinylated anti-GP polyclonal detector antibody (Proteintech Cat# 24494-1-AP, RRID:AB_2879573). After washing steps, a conjugate of streptavidin-β-galactosidase (SβG) was added to label the captured GP. After additional washing, resorufin β-D-galactopyranoside (RGP, i.e., the substrate for SBG) was added, and the labeled beads were transferred to the Simoa array. Fluorescent signal was captured and quantified by the SR-X instrument as average enzymes per bead (AEB), and then the GP concentration of each unknown lysate was interpolated from the GP standard curve.

### Animals, drug treatments, and phenotypic analyses

All procedures involving mice were in compliance with the ARRIVE guidelines and approved by the Institutional Animal Care and Use Committee (IACUC) at Thomas Jefferson University. All animals were housed in standard cages and provided food and water ad libitum in a temperature-, light-, and humidity-controlled animal facility. Six-month-old Tg(C9orf72_i3)112Lutzy (Jackson Laboratory) and wild-type littermate mice were administered i.p. injections of saline (vehicle) or 2 g/kg 2DG (Cayman Chemical). For acute studies, animals were administered either a single dose (2 g/kg 2DG), or higher doses (4–8 g/kg 2DG), which were achieved by repeated injections with 2 g/kg 2DG each spaced out by 1–2 h. For chronic studies, animals were injected twice weekly with 2 g/kg 2DG for six weeks. For all studies, animals were euthanized by $CO_2$ inhalation 16 h following the final injection and transcardially perfused with ice-cold PBS. Whole brains and spinal cords were then excised, flash frozen in liquid $N_2$, and stored at –80 °C for biochemical analysis.

For inverted wire hang motor assessments, animals were suspended upside-down from a wire mesh and their latency to fall (capped at 180 s) was recorded. Each animal was given 3–4 independent trials with >10 min of rest between each trial. Baseline assessments were acquired several days before the first 2DG injection; subsequent assessments were acquired weekly and on a different day than the 2DG injections. The average of the 3 trials for each animal at each time-point was used for analysis. All test sessions were performed in the light cycle phase (10 am–5 pm).

For confocal scanning laser ophthalmoscopy (cSLO) imaging, mice were anesthetized with ketamine (100 mg/kg) and xylazine (10 mg/kg) and eyes were anesthetized with 0.5% proparacaine HCl ophthalmic solution (NDC: 17478-263-12, Akorn). Pupils were dilated with 1% tropicamide eye drops (NDC: 17478-102-12, Akorn). Ocular eye shields and Systane Ultra Lubricant Eye Drops (Alcon Laboratories) were used to keep eyes hydrated. cSLO images were obtained using a Spectralis HRA + OCT (Heidelberg Engineering). Mice were positioned with the optic nerve in the center of the image using a 55° field of view (FOV) lens and imaged with two different modes, infrared (IR) and blue autofluorescence (BAF). All images were acquired with the auto-normalization activated, which provided the best contrast.

### Statistical analyses

All statistical analyses were performed using GraphPad Prism 9.0/10.0—these include Kaplan–Meier survival assessment with log-rank testing (to compare survival curves), student's two-tailed t-test (to compare the means of two groups), two-tailed t-test with Welch's correction (to compare the means of two groups with unequal variances or sample sizes), one-way analysis of variance (ANOVA) with Dunnett's multiple comparisons test or two-way ANOVA (to compare the means of more than two groups). A minimum of 30 individual cells were analyzed per condition for single-cell analyses. Data are reported as mean ± SEM. $p$-value < 0.05 was considered statistically significant.

The experimenter was blinded to the genotype of animals during behavioral data acquisition and unblinded before analysis. No behavioral data points were excluded from the analysis.

## Data availability

The RNA-seq and metabolomics data produced in this study have been uploaded to Gene Expression Omnibus (accession number GSE260665) and Metabolomics workbench (datatrack_id:4680; study_id: ST003112). The sequences for all DNA plasmids used in this study are uploaded to GenBank (https://www.ncbi.nlm.nih.gov/genbank/) with the following accession numbers and respective URLs: BankIt2694892 pLV_hSyn_-GA188_EGFP OQ828708 (https://www.ncbi.nlm.nih.gov/nuccore/OQ828708), BankIt2694894 pLV_hSyn_GFP OQ828709 (https://www.ncbi.nlm.nih.gov/nuccore/OQ828709.1/), BankIt2694876 pLV_hSyn_-GA50_GFP OQ828707 (https://www.ncbi.nlm.nih.gov/nuccore/OQ828707), BankIt2694895 pLV_hSyn_GR50_GFP OQ828710 (https://www.ncbi.nlm.nih.gov/nuccore/OQ828710), BankIt2694896 pLV_hSyn_PA50_GFP OQ828711 (https://www.ncbi.nlm.nih.gov/nuccore/OQ828711), BankIt2694899 pLV_hSyn_PR50_GFP OQ828712 (https://www.ncbi.nlm.nih.gov/nuccore/OQ828712).

The source data of this paper are collected in the following database record: biostudies:S-SCDT-10_1038-S44319-024-00140-7.

## Peer review information

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

## Acknowledgements

We thank Target ALS for providing the anti-GP antibody (cat # TALS 828.179) and Drs. M.E. Ward and S.J. Barmada for providing iPSC lines. We also thank the members of the Weinberg ALS Center for providing critical feedback throughout the development of this project. The following sources supported this work: National Institutes of Health F31-NS118838 (ATN), R21-NS090912 (DT), RF1-AG057882 (DT), RF1-NS114128 (AH), and R01-NS109150 (PP); Muscular Dystrophy Association grant 628389 (DT); DoD grant W81XWH-21-1-0134 (MEC); Family Strong for ALS, Farber Family Foundation, and the Aldrich Foundation (Weinberg ALS Center). The Weill Cornell Medicine Meyer Cancer Center Proteomics & Metabolomics Core Facility for performing and analyzing mass spectrometry.

## Author contributions

**Andrew T Nelson**: Conceptualization; Data curation; Formal analysis; Funding acquisition; Investigation; Methodology; Writing—original draft; Writing—review and editing. **Maria Elena Cicardi**: Conceptualization; Data curation; Formal analysis; Methodology; Writing—original draft; Writing—review and editing. **Shashirekha S Markandaiah**: Investigation. **John YS Han**: Investigation. **Nancy J Philp**: Supervision. **Emily Welebob**: Formal analysis. **Aaron R Haeusler**: Supervision. **Piera Pasinelli**: Supervision; Project administration. **Giovanni Manfredi**: Supervision. **Hibiki Kawamata**: Investigation. **Davide Trotti**: Conceptualization; Data curation; Supervision; Funding acquisition; Writing—original draft; Project administration; Writing—review and editing.

Source data underlying figure panels in this paper may have individual authorship assigned. Where available, figure panel/source data authorship is listed in the following database record: biostudies:S-SCDT-10_1038-S44319-024-00140-7.

## Disclosure and competing interests statement

The authors declare no competing interests.

# Expanded View Figures

**Figure EV1.   Analysis of brain metabolites of C9-BAC vs. WT mice.**

(**A**) Schematic depicting the bacterial artificial chromosome (BAC) transgene used to drive expression of the entire human *C9orf72* gene with a (GGGGCC)$_{100\text{-}1,000}$ repeat expansion in intron 1 (C9-BAC). Adapted from O'Rourke et al (2015). (**B**) LC-MS measurement of relative glucose concentrations in the frontal cortex of C9-BAC versus WT animals. (**C**) Schematic depicting the tricarboxylic acid (TCA) cycle pathway with all metabolic intermediates. LC-MS measurement of relative α-ketoglutarate concentrations in the frontal cortex of C9-BAC animals versus WT animals. (**D**) LC-MS measurement of the relative concentrations of four amino acids (isoleucine, cysteine, lysine, and ornithine) in the frontal cortex of C9-BAC vs. WT animals. (**E**) LC-MS measurement of relative NADP +, NADH, and NADPH concentrations in the frontal cortex of C9-BAC vs WT. All individual metabolite data are shown as median-normalized and log-transformed values (abbreviated as "Normalized conc."). For box and whisker plots, box edges denote upper and lower quartiles, horizontal lines within each box denote median values, whiskers denote maximum and minimum values, and shaded circles denote individual values for each animal. Student's two-tailed t-test, *$p < 0.05$, **$p < 0.01$. Source data are available online for this figure.

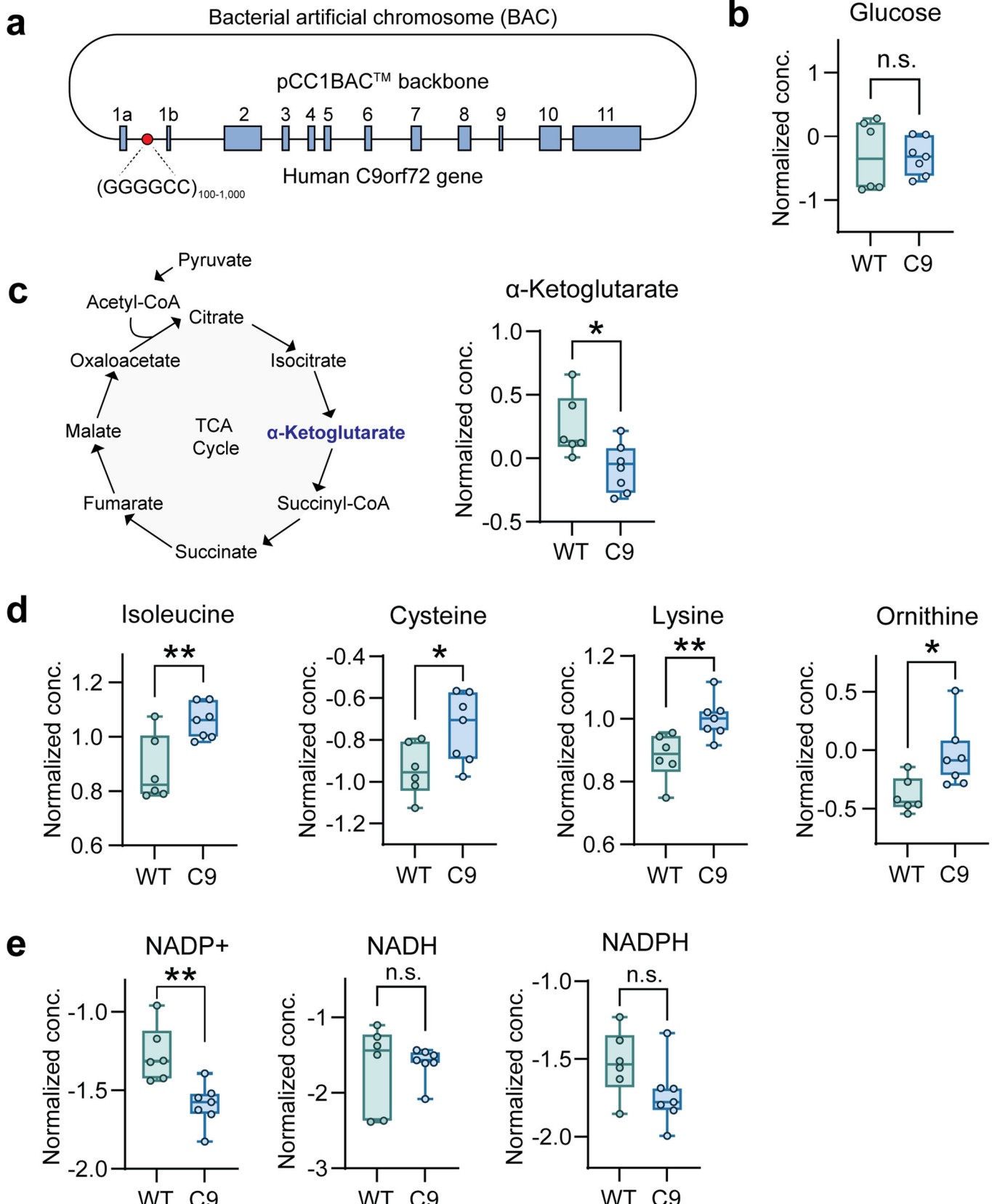

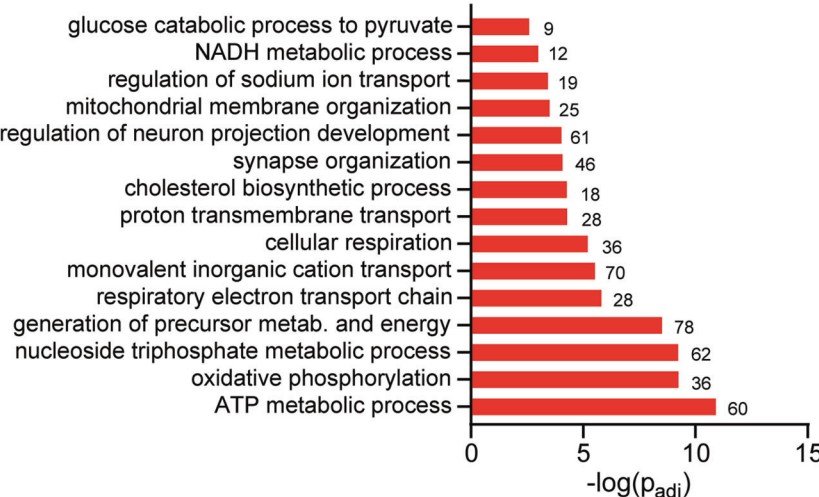

**a** Enriched GO pathways (*decreased* in 2DG-treated)

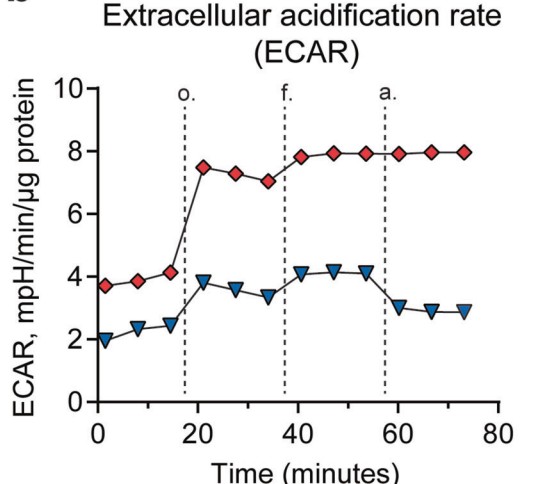

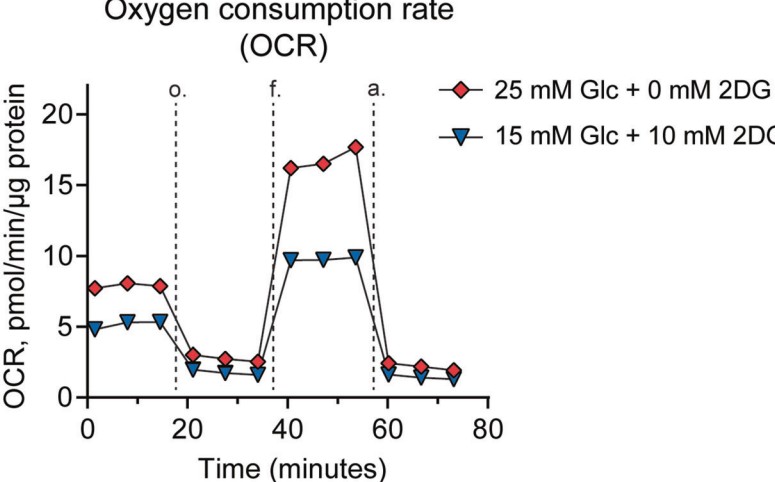

Figure EV2. Effect of 2DG on cellular respiration.

(A) RNA sequencing measurement of select downregulated gene ontology (GO) pathways in C9orf72 patient-derived i³Neurons incubated with 10 mM 2DG-containing media versus normo-glucose media for 48 h ($n = 2$ individual i³Neuron lines with 2 separate differentiations per line). (B) Seahorse extracellular flux assay-based measurement of extracellular acidification rate (ECAR) normalized to total protein and oxygen consumption rate (OCR) normalized to total protein in primary neurons incubated with either normo-glucose media or 10 mM 2DG-containing media for 48 h. Cells were sequentially treated with 1.5 µg/ml oligomycin (o.), 3 µM FCCP (f.), and 1 µM antimycin (a.) during the assay ($n = 1$ with 3 technical replicates). For (A), the values adjacent to each bar represent the number of altered genes in each GO pathway. The statistical test applied is the Fischer's exact test with a Benjamini–Hochberg False Discovery Rate (FDR). For (B) data are presented as mean ± SEM of assay technical replicates. Source data are available online for this figure.

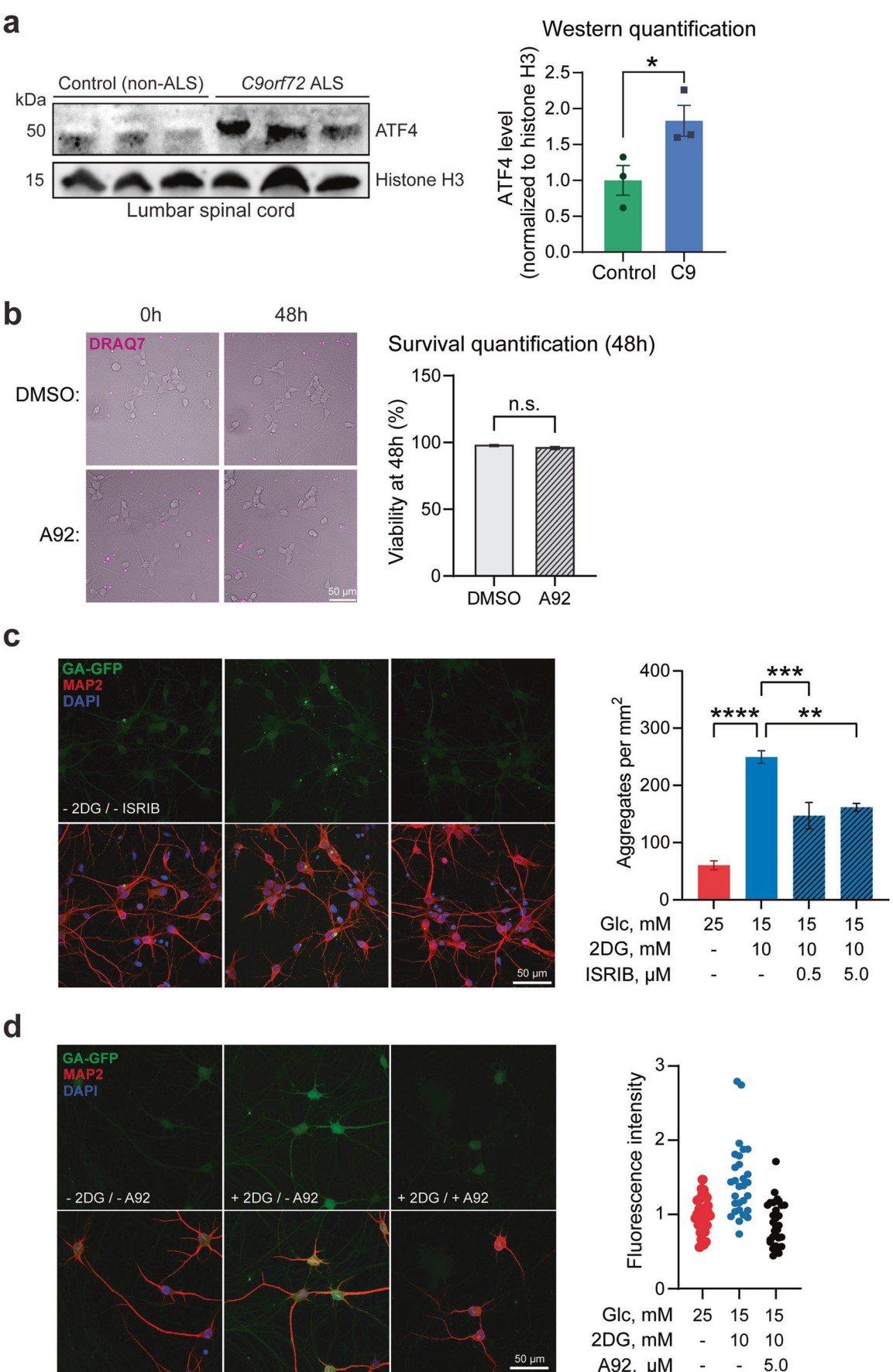

◀ **Figure EV3. Inhibition of ISR rescues DPR aggregation.**

(A) Western blot analysis and quantification of ATF4 expression level relative to histone H3 expression level (as a loading control) in lumbar spinal cord tissue homogenates from either healthy control subjects or C9orf72-ALS patients ($n = 3$ subjects per genotype). (B) Live-cell imaging of human i³Neurons treated with either 0.1% DMSO or 5.0 µM A92 and 0.1% DMSO for 48 h, with corresponding quantification of survival proportions. DRAQ7 was used as a fluorescent dead cell indicator ($n = 2$ i³Neuron lines with 2 independent differentiations per line). Neurons were imaged live across 5 randomly selected fields of view/conditions. The same field of view was imaged daily. More than 100 neurons were tracked/condition/experiment. (C) Fluorescent confocal imaging and quantification of DPR aggregate formation in primary neurons transduced with RAN translation vector, then incubated with either normo-glucose media, 10 mM 2DG-containing media, or 10 mM 2DG-containing media with 0.5 µM or 5 µM ISRIB (all in the presence of 0.1% DMSO; $n = 4$). (D) Fluorescent confocal imaging and quantification of DPR formation in human i³Neurons transduced with RAN translation vector, then incubated with either normo-glucose media, 10 mM 2DG-containing media, or 10 mM 2DG-containing media with 2.5 µM A92 (all in the presence of 0.1% DMSO; $n = 1$). At least 30 neurons were analyzed/condition. For (A, B), student's two-tailed t-test. For (C), one-way ANOVA with Dunnett's test for multiple comparisons. All data are presented as mean ± SEM. $*p < 0.05$, $**p < 0.01$, $***p < 0.001$, $****p < 0.0001$. Source data are available online for this figure.

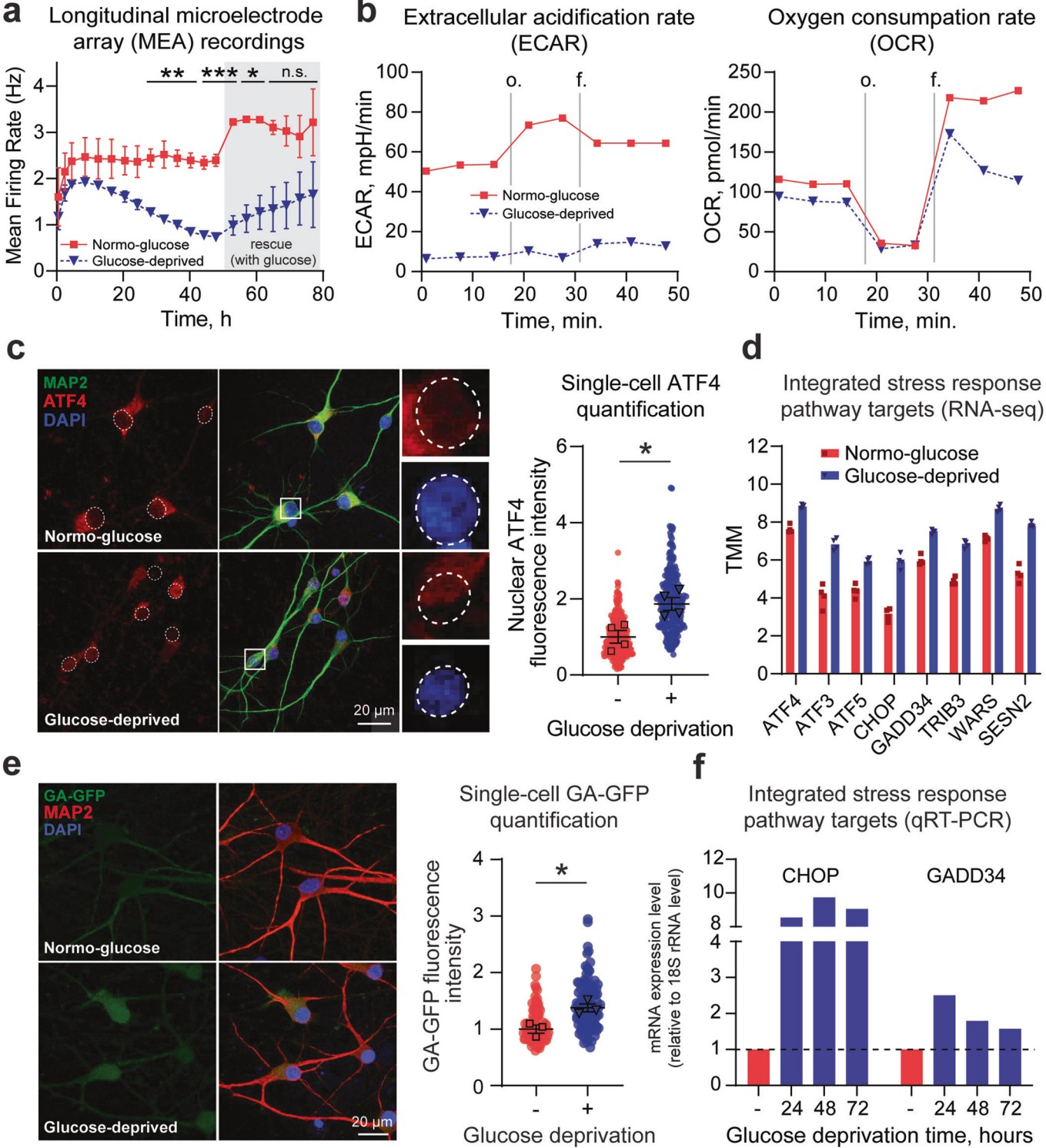

**Figure EV4.  Effect of glucose deprivation in I³neurons.**

(A) Longitudinal microelectrode array (MEA)-based measurement of spontaneous neuronal activity of i³Neurons (healthy control line) incubated in either normo-glucose or glucose-deprived media over a 48-h period ($n = 3$ biological replicates). Glucose deprivation was initiated at time $= 0$ h. (B) Seahorse extracellular flux assay-based measurement of extracellular acidification rate (ECAR) normalized to total protein and oxygen consumption rate (OCR) normalized to total protein of primary neurons immediately following incubation with either normo-glucose media or glucose-deprived media for 48 h ($n = 1$ with 3 technical replicates). Cells were sequentially treated 1.5 µg/ml oligomycin (o.) and 3 µM FCCP (f.) during the assay. (C) Immunofluorescence-based measurement and quantification of nuclear ATF4 expression level in MAP2-positive primary neurons immediately following incubation with either normo-glucose media or glucose-deprived media for 48 h ($n = 4$ biological replicates). Neurons were imaged live across >5 randomly selected fields of view/conditions. The same field of view was imaged daily. More than 100 neurons were tracked/condition/experiment. (D) RNA sequencing analysis of select individual ISR target transcripts (all $p_{adj} < 0.05$) in i³Neurons immediately following exposure to either normo-glucose or glucose-deprived media for 48 h ($n = 2$ i³Neuron lines with 2 differentiations per line). (E) Representative images of i³Neurons transduced with RAN translation vector and then cultured in either normo-glucose or glucose-deprived media for 48 h, with corresponding quantification of GA-GFP fluorescence intensity ($n = 3$ biological replicates). Neurons were imaged live across 5 randomly selected fields of view/conditions. The same field of view was imaged daily. More than 100 neurons were tracked/condition/experiment. (F) qRT-PCR measurement of mRNA levels of two ISR transcriptional targets (CHOP and GADD34) in i³Neurons immediately following incubation with either normo-glucose or glucose-deprived media for either 24, 48, or 72 h ($n = 1$). For (A), multiple student's two-tailed t-tests. For (C–E), student's two-tailed t-test. All data (except D and F) are presented as mean ± SEM. *$p < 0.05$, **$p < 0.01$, ***$p < 0.001$. Source data are available online for this figure.

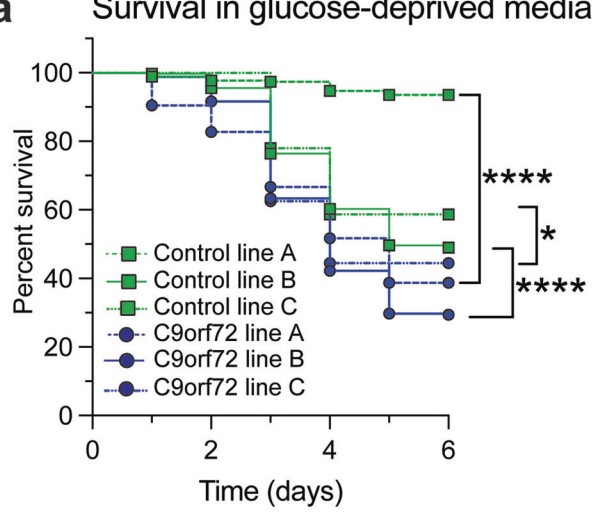

**a** Survival in glucose-deprived media

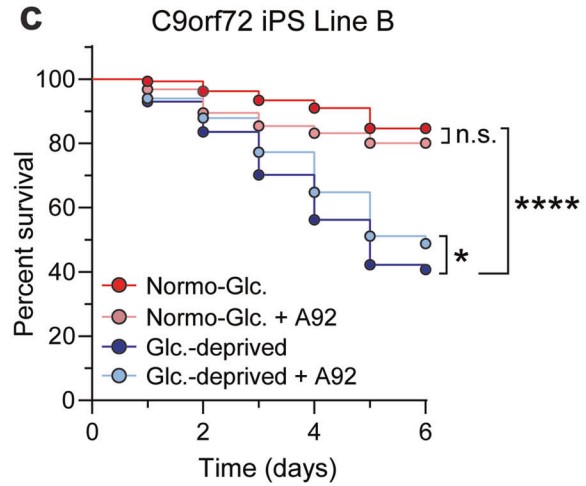

**c** C9orf72 iPS Line B

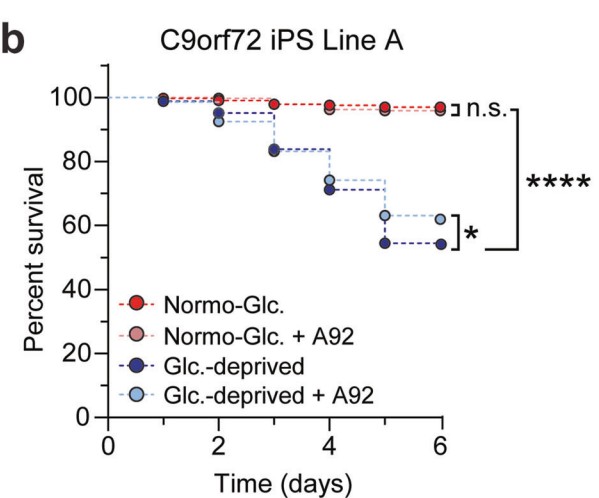

**b** C9orf72 iPS Line A

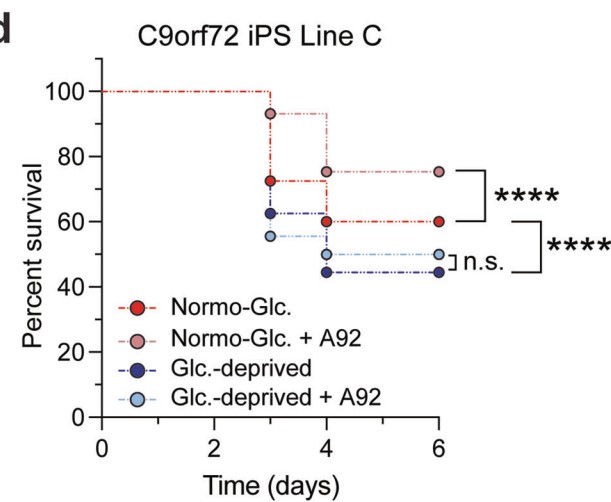

**d** C9orf72 iPS Line C

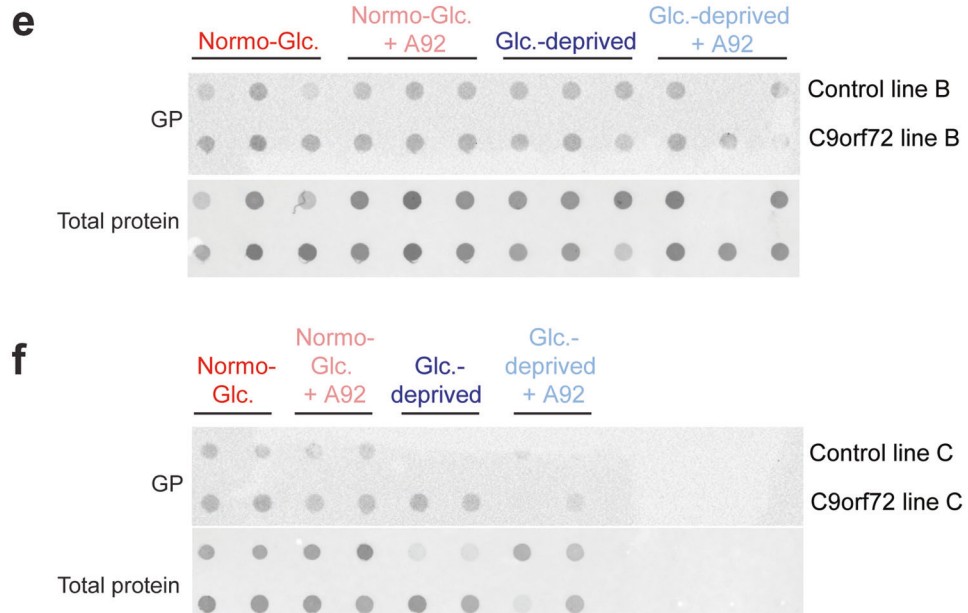

**Figure EV5. Glucose deprivation in control and C9orf72 derived I³neurons.**

(A) Kaplan–Meier survival analysis of i³Neurons derived from either C9orf72 patients or healthy controls and maintained in glucose-deprived media ($n = 3$ i³Neuron lines per genotype with 3 independent differentiations per line). Line A, B, C. ***$p$<. (B) Kaplan–Meier survival analysis of C9orf72 patient-derived i³Neurons maintained in either normo-glucose or glucose-deprived media and treated with either 2.5 µM A92 with 0.1% DMSO or 0.1% DMSO only as vehicle control (3 independent differentiations per line). Line A. (C) Kaplan–Meier survival analysis of C9orf72 patient-derived i³Neurons maintained in either normo-glucose or glucose-deprived media and treated with either 2.5 µM A92 with 0.1% DMSO or 0.1% DMSO only as vehicle control (3 independent differentiations per line). Line B. (D) Kaplan–Meier survival analysis of C9orf72 patient-derived i³Neurons maintained in either normo-glucose or glucose-deprived media and treated with either 2.5 µM A92 with 0.1% DMSO or 0.1% DMSO only as vehicle control (3 independent differentiations per line). Line C. (E, F). Dot blot assessment of 8 M urea-soluble GP levels and relative total protein levels in i³Neurons derived from either C9orf72 patients or healthy controls maintained in either normo-glucose or glucose-deprived media and treated with either 2.5 µM A92 with 0.1% DMSO or 0.1% DMSO only as vehicle control. ($n =$ two independent lines per genotype with 2 or 3 technical replicates per treatment) Kaplan–Meier log-rank survival test. *$p < 0.05$, **$p < 0.01$, ****$p < 0.0001$. Neurons were imaged live across >5 randomly selected fields of view/conditions. The same field of view was imaged daily. More than 100 neurons were tracked/condition/experiment. All data are presented as mean ± SEM ($n = 3$ biological replicates). *$p < 0.05$, **$p < 0.01$. Source data are available online for this figure.

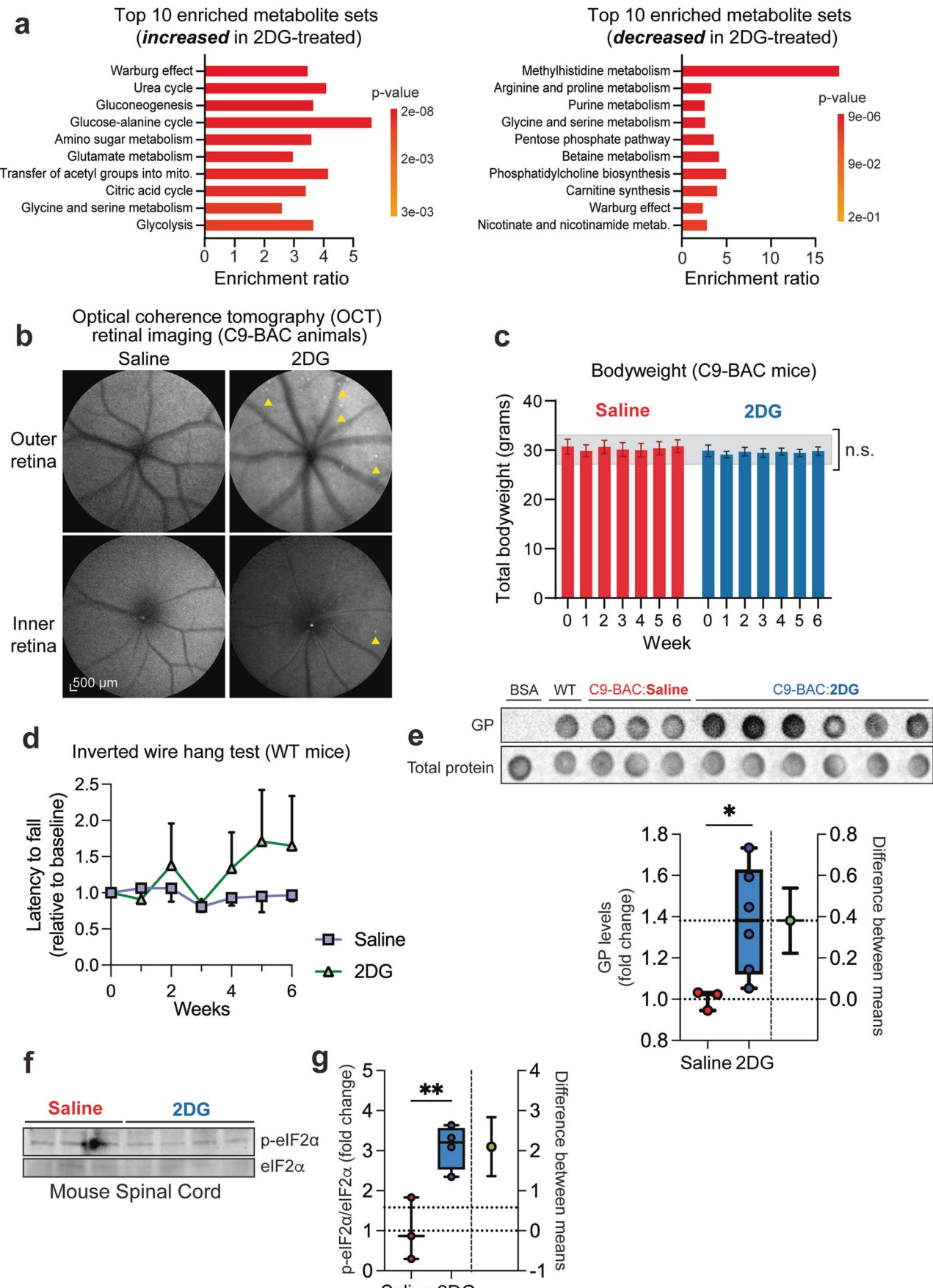

**Figure EV6.  Effect of 2DG treatment in rodents.**

(A) Enrichment analysis of significantly altered metabolite sets in the frontal cortex of C9-BAC animals immediately following chronic exposure to either 4 g/kg/week 2DG or saline ($n = 7$ animals per condition). Enrichment ratio represents the number of metabolites within each metabolite set that are either increased (on the left) or decreased (on the right). (B) Optical coherence tomography (OCT) retinal imaging of C9-BAC animals immediately following chronic exposure to either 4 g/kg/week 2DG or saline. Yellow arrows indicate hyperreflective foci ($n = 2$–3 animals per condition). (C) Longitudinal measurement of body weight during chronic exposure of C9-BAC animals to 4 g/kg/week 2DG or saline ($n = 7$ animals per condition). (D). Longitudinal assessment of inverted wire hang performance of wild-type littermate control animals chronically exposed to 4 g/kg/week 2DG ($n = 2$ animals) or saline ($n = 3$ animals). (E) Dot blot assessment and corresponding quantification of 8 M urea-soluble GP levels relative to total protein levels in the spinal cord C9-BAC animals treated with either saline ($n = 3$ animals) or 2DG ($n = 6$ animals). Bovine serum albumin (BSA) and spinal cord lysate from a wild-type animal were used as controls. (F) Western Blot analysis for p-eIF2α and eIF2α of mouse brain cortices treated with saline or 2DG. (G) Quantification of the ratio between pEif2α and Eif2α protein of the WB in Fig EV6f ($n = 3$ biological replicates). For (A), the statistical test applied is the Fischer's exact test with a Benjamini–Hochberg False Discovery Rate (FDR). For (C, D), two-way ANOVA. For (E–G), two-tailed t-test with Welch's correction. All data (except A) are presented as mean ± SEM. *$p < 0.05$, n.s. $p > 0.05$. For box and whisker plots, box edges denote upper and lower quartiles, horizontal lines within each box denote median values, whiskers denote maximum and minimum values, and shaded circles denote individual values for each replicate. Source data are available online for this figure.

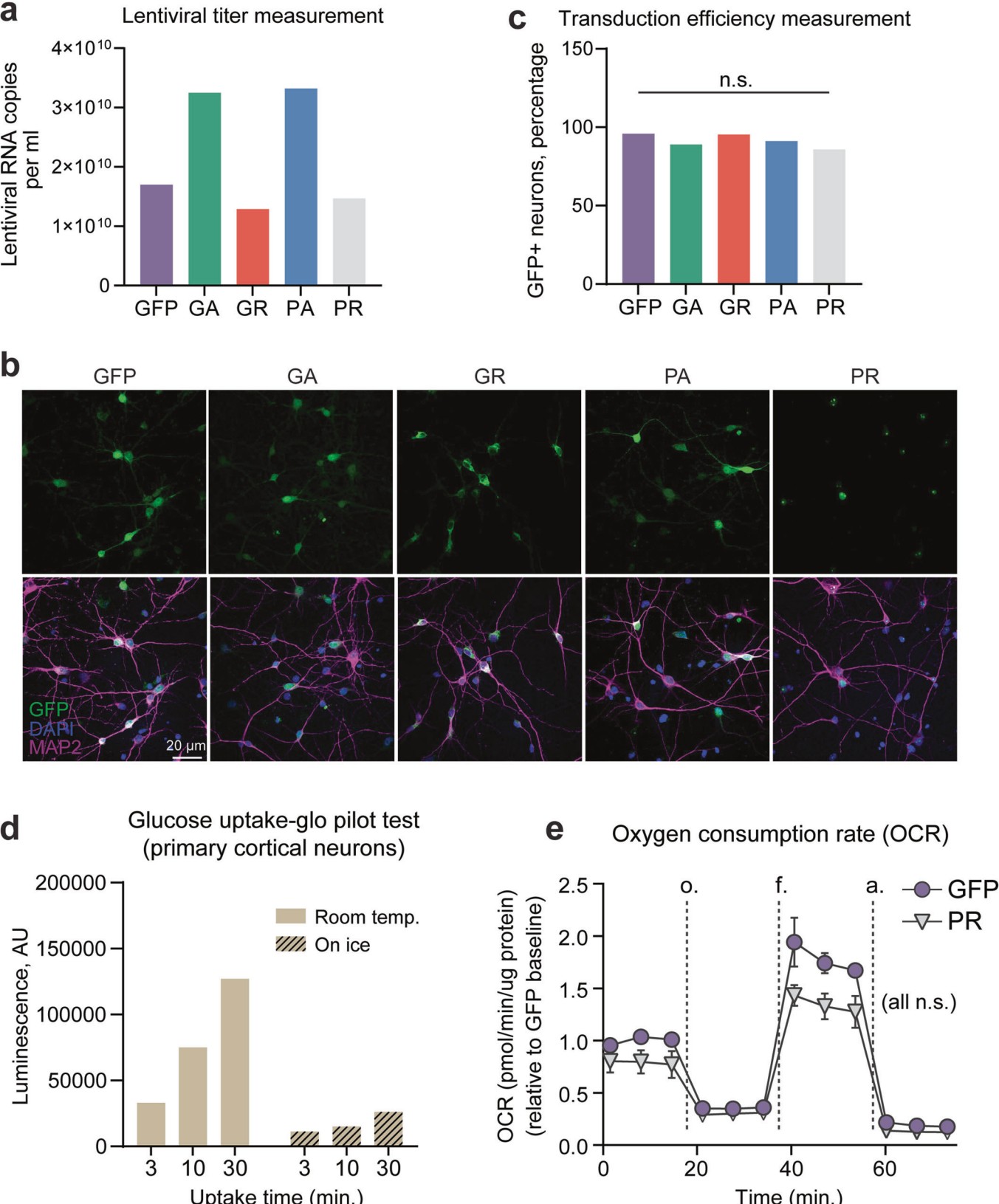

**a** Lentiviral titer measurement

**c** Transduction efficiency measurement

**b** GFP GA GR PA PR

GFP
DAPI
MAP2
20 μm

**d** Glucose uptake-glo pilot test (primary cortical neurons)

**e** Oxygen consumption rate (OCR)

**Figure EV7. DPRs transduction in rat cortical neurons.**

(A) qRT-PCR measurement of DPR and GFP-only vector lentiviral titers ($n = 1$ technical replicate). (B) Representative images of primary neurons transduced with DPR or GFP-only lentiviral vectors, then stained for MAP2. (C) Quantification of the percentage of MAP2-positive cells also positive for GFP from (B) ($n = 1$ with 3 technical replicates). (D) Validation of luminescent assay (time- and temperature dependence) for measurement of glucose uptake in primary neurons ($n = 1$ with 3 technical replicates). (E) Seahorse extracellular flux assay measurement of oxygen consumption rate (OCR normalized to total protein) of primary neurons transduced with either the PR or GFP-only vector ($n = 3$ biological replicates). Cells were sequentially treated with 1.5 µg/ml oligomycin (o.), 3 µM FCCP (f.), and 1 µM antimycin (a.) during the assay. All data (except A) are presented as mean ± SEM. For (C), one-way ANOVA with Dunnett's multiple comparisons tests. For (E), multiple student's two-tailed t-test. n.s. $p > 0.05$. Source data are available online for this figure.

