## [Peer Review File · EMBO Reports]

Glucose Hypometabolism Prompts RAN Translation and Exacerbates C9orf72-related ALS/FTD Phenotypes

Andrew T. Nelson, Maria Elena Cicardi, Shashirekha S. Markandaiah, John Y.S. Han, Nancy J. Philp, Emily Welebob, Aaron R. Haeusler, Piera Pasinelli, Giovanni Manfredi, Hibiki Kawamata and Davide Trotti

Corresponding author(s): Davide Trotti (davide.trotti@jefferson.edu)

Review Timeline:

Submission Date:	27th Jun 23
Editorial Decision:	14th Aug 23
Revision Received:	26th Feb 24
Editorial Decision:	25th Mar 24
Revision Received:	4th Apr 24
Accepted:	9th Apr 24

Editor: Esther Schnapp

Transaction Report:

Dear Prof. Trotti,

Thank you for the submission of your manuscript to EMBO reports. We have now received the comments from 2 referees. I could not secure a third referee for your study, and think that the 2 reports we have are sufficient to make a decision.

As you will see, the referees acknowledge that the findings are interesting. However, they also point out that the data need to be strengthened to support the main conclusions. I think all suggestions are good and should be addressed. Please let me know in case you disagree and we can discuss the revision requirements further, also in a video chat, if you like.

I would thus like to invite you to revise your manuscript with the understanding that the referee concerns must be fully addressed and their suggestions taken on board. Please address all referee concerns in a complete point-by-point response. Acceptance of the manuscript will depend on a positive outcome of a second round of review. It is EMBO reports policy to allow a single round of major revision only and acceptance or rejection of the manuscript will therefore depend on the completeness of your responses included in the next, final version of the manuscript.

We realize that it is difficult to revise to a specific deadline. In the interest of protecting the conceptual advance provided by the work, we recommend a revision within 3 months (14th Nov 2023). Please discuss the revision progress ahead of this time with the editor if you require more time to complete the revisions.

- 1) A data availability section providing access to data deposited in public databases is missing. If you have not deposited any data, please add a sentence to the data availability section that explains that.
- 2) Your manuscript contains statistics and error bars based on $n=2$. Please use scatter blots in these cases. No statistics should be calculated if $n=2$.

5) a complete author checklist, which you can download from our author guidelines <https://www.embopress.org/page/journal/14693178/authorguide>. Please insert information in the checklist that is also reflected in the manuscript. The completed author checklist will also be part of the RPF.

6) Please note that all corresponding authors are required to supply an ORCID ID for their name upon submission of a revised manuscript (<https://orcid.org/>). Please find instructions on how to link your ORCID ID to your account in our manuscript tracking system in our Author guidelines <https://www.embopress.org/page/journal/14693178/authorguide#authorshipguidelines>

7) Before submitting your revision, primary datasets produced in this study need to be deposited in an appropriate public

database (see <https://www.embopress.org/page/journal/14693178/authorguide#datadeposition>). Please remember to provide a reviewer password if the datasets are not yet public. The accession numbers and database should be listed in a formal "Data Availability" section placed after Materials & Method (see also <https://www.embopress.org/page/journal/14693178/authorguide#datadeposition>). Please note that the Data Availability Section is restricted to new primary data that are part of this study. * Note - All links should resolve to a page where the data can be accessed. *

10) Regarding data quantification (see Figure Legends:

<https://www.embopress.org/page/journal/14693178/authorguide#figureformat>)

I look forward to seeing a revised form of your manuscript when it is ready.

Yours sincerely,

Referee #1:

In this manuscript by Nelson and colleagues, they explore the relationship between glucose hypometabolism and RAN translation in the pathogenesis of C9orf72 associated FTD and ALS. They see that C9 BAC mice have metabolic signatures that correlate with glucose hypometabolism. They then demonstrate that treatment with 2 Deoxyglucose- which effectively creates a hypometabolic state- is sufficient to boost RAN translation of G4C2 repeats (at least in the GA reading frame from linear reporters) through activation of GCN2 and eIF2alpha phosphorylation. They then show that 2DG is bad for patient iPSCs and the C9-BAC mice. Lastly, they show that overexpression of PR DPRs interferes with glucose metabolism in neurons, suggesting a potential mechanism by which the hypometabolism occurs in the first place.

Overall, this is an interesting manuscript with a significant body of data. I think their hypothesis is intriguing. Figures 1 thru 4 are well laid out and add significant new information to the field. Both figure 5 and 6 support their hypothesis but both have some significant limitations. Figure 5 shows only very modest beneficial effects in a single phenotypic readout in 2 iPSC lines (Figure 5). Figure 6 is hard to interpret given the existing hypometabolism present in these mice (that they demonstrate) and the logical flaw with doing two separate bad things (mutant gene and further lowering glucose) and assuming they are worse together because of some interactivity of these pathways. Figure 7 provides some evidence for how PR could play a role in this process, but interpretation of this data is also somewhat problematic given that PR elicits a host of toxic cascades, of which altered glucose metabolism may be just one. These last 3 figures leave the central premise of the manuscript - that RAN translation is both triggered by and triggers GCN2 dependent stress cascades that contribute to ALS pathogenesis- largely still untested. I think fully testing this hypothesis (e.g. by assessing whether loss of GCN or any of the drugs they spell out in the first or second paragraph of their discussion are sufficient to actually rescue disease-relevant phenotypes in the C9 iNeurons or C9 BAC mice) extends beyond the scope of this manuscript. As such, while the manuscript could be significantly improved by additional experiments, I think that most of the issues I observe can be addressed with alterations in the language to acknowledge the limitations of the data presented.

Specific suggestions:

1) Abstract: Change to

" These findings provide a potential mechanistic link between energy imbalance and C9-ALS/FTD pathogenesis and suggest a feedforward loop model that opens several opportunities for therapeutic intervention."

2) Line 296:

In this study, we identified a novel link between energy imbalance and disease pathogenesis in models of C9orf72-ALS/FTD, which potentially reveals some opportunities for therapeutic intervention.

3) Lines 302-321: There is a lot of speculation on potential therapeutic interventions in patients here that are not warranted by the data in the manuscript. Recommending therapies or even clinical trials based on the data presented in this manuscript is premature and should be significantly limited.

4) Line 321:

Overall, our findings warrant additional preclinical efforts to determine the impact of preserving bioenergetic homeostasis in C9orf72 ALS/FTD model systems - in both symptomatic and presymptomatic conditions in the hopes that such approaches might reveal potential therapies that can be tested in C9orf72 ALS/FTD patients.

5) Line 341: This is too speculative on ISRIB.

6) The authors should include a paragraph that addresses the limitations of the studies performed (see paragraph 2, this review) and provide future directions or approaches that might address these.

Referee #2:

C9ORF72 hexanucleotide repeat expansion is the most common genetic cause of ALS and FTD. In this manuscript, the authors reported disruption of energy and amino acid production in the C9-BAC mice. They showed that the glucose hypometabolism leads to increased RAN translation of the repeat RNA by GCN2 kinase-mediated integrated stress response (ISR) using reporter cells. They further showed GCN2 inhibition can rescue the disease phenotypes of patient iPSC-neurons, and glycolysis inhibition can increase poly-dipeptides and enhance phenotypes in C9-BAC mice. It has been reported that RAN translation can be increased by ISR. GCN2 and hypometabolism is one of the upstream factors that trigger ISR. It is an interesting finding that this branch also contributes to DPR toxicity and disease phenotypes, using both in vivo and in vitro models. Some of the conclusions need more evidence to support before publication.

1. The activation of eIF2a pathway should be measured using phosphor-eIF2a antibody. Only indirect evidence (such as ATF4) was used in the whole manuscript.

2. The effect on endogenous DPR levels need to be measured in patient derived neurons. Especially, the authors reasoned that the neuron survival changes is due to the DPR reduction. More data need to be provided to support this claim.

3. In Figure 5d, there is big variation of cell response to glucose deprivation. And the rescue effect is small in Figure 5e. More iPSC cell lines and replicates are needed to support the conclusion.

4. The DPR levels need to be measured using more quantitative methods, such as ELISA, in both cells and mice. The dot blot data are not very convincing.

5. Overall, the statistics need to be improved. It is often not clear (or missing) the key information about replicates, cell numbers that were quantified.

6. The authors should modify the writing and avoid overclaim. For example, the worsened phenotypes by 2DG treatment correlates with DPR change (which also need to be confirmed using better method). But this evidence itself does not prove the causal relationship. In particular, the timeline (which need better description) of DPR and behavior changes seems not supporting DPR is the only cause.

We appreciate the valuable suggestions made by the referees and agree that they should be considered during the revision process. We have provided a detailed response to all of the referee's concerns, addressing each comment thoroughly and transparently. We have included in the revised manuscript additional data, the necessary clarifications, and modifications as required. Below, is our point-by-point response to the reviewers' comments.

Comments Referee #1: In this manuscript by Nelson and colleagues, they explore the relationship between glucose hypometabolism and RAN translation in the pathogenesis of C9orf72-associated FTD and ALS. They see that C9 BAC mice have metabolic signatures that correlate with glucose hypometabolism. They then demonstrate that treatment with 2 Deoxyglucose- which effectively creates a hypometabolic state- is sufficient to boost RAN translation of G4C2 repeats (at least in the GA reading frame from linear reporters) through activation of GCN2 and eIF2alpha phosphorylation. They then show that 2DG is bad for patient iPSCs and the C9-BAC mice. Lastly, they show that overexpression of PR DPRs interferes with glucose metabolism in neurons, suggesting a potential mechanism by which the hypometabolism occurs in the first place. Overall, this is an interesting manuscript with a significant body of data. I think their hypothesis is intriguing. Figures 1 thru 4 are well laid out and add significant new information to the field. Both figure 5 and 6 support their hypothesis but both have some significant limitations.

Figure 5 shows only very modest beneficial effects in a single phenotypic readout in 2 iPSC lines (Figure 5).

Response: We have added a third C9 iPSC line to our experiments. In Fig 5, we present now the data for all three lines together. Moreover, we have provided a breakdown of the survival analysis for each iPSC line separately in Fig. EV5 for transparency and scientific rigor.

Figure 6 is hard to interpret given the existing hypometabolism present in these mice (that they demonstrate) and the logical flaw with doing two separate bad things (mutant gene and further lowering glucose) and assuming they are worse together because of some interactivity of these pathways.

Response: In response to the criticism of Figure 6, it's understandable that interpreting the data might be challenging given the pre-existing hypometabolism in the C9 BAC mice. However, we presented compelling evidence that the combination of the human mutant hexanucleotide repeat expansion and further lowering glucose levels exacerbates the phenotype beyond what would be expected by the individual effects alone. This suggests an interaction between the pathways involved, which is a common phenomenon in complex biological systems. While it's true that both factors individually lead to negative outcomes, the synergistic effect observed in Figure 6 underscores the importance of understanding the interplay between glucose metabolism and RAN translation in the context of C9orf72-associated ALS and FTD. We agree that further studies elucidating the molecular mechanisms underlying this interaction would strengthen the conclusions drawn from this intriguing manuscript. These additional studies are complex and underway. However, we believe they are beyond the scope and depth of this initial report.

The translational aspect of human disease is crucial in interpreting findings from animal models like the C9orf72 ALS mouse model, which has a shorter lifespan than humans. In response to the concern about the relevance of exacerbating conditions in the mouse model, we believe it's important to acknowledge that while the mouse model may not fully recapitulate the human disease's progression and lifespan, it provides valuable insights into disease mechanisms and potential therapeutic interventions. Without exacerbating conditions such as further lowering glucose levels, the mice would be expected to still exhibit the characteristic features of C9orf72 ALS, albeit at a slower rate or to a lesser degree. The presence of hypometabolism in the brains

of these mice indicates that metabolic dysfunction is already a significant aspect of the disease pathology. However, by exacerbating this condition, we aimed to mimic a scenario where metabolic dysfunction is further compromised, potentially accelerating disease progression or exacerbating specific aspects of the pathology. In studying human disease in a mouse model with a limited lifespan, exacerbating conditions can provide valuable insights into potential disease mechanisms and therapeutic targets within a feasible timeframe. While the exact extrapolation to human patients should be done cautiously, understanding the impact of exacerbating conditions in the mouse model can inform future studies and therapeutic strategies to mitigate metabolic dysfunction in C9orf72 patients. Therefore, despite its limitations, we firmly believe that the findings shown in Figure 6 contribute to our understanding of disease pathology and potential therapeutic avenues for further investigation.

Figure 7 provides some evidence for how PR could play a role in this process, but interpretation of this data is also somewhat problematic given that PR elicits a host of toxic cascades, of which altered glucose metabolism may be just one. These last 3 figures leave the central premise of the manuscript - that RAN translation is both triggered by and triggers GCN2 dependent stress cascades that contribute to ALS pathogenesis- largely still untested. I think fully testing this hypothesis (e.g. by assessing whether loss of GCN or any of the drugs they spell out in the first or second paragraph of their discussion are sufficient to actually rescue disease-relevant phenotypes in the C9 iNeurons or C9 BAC mice) extends beyond the scope of this manuscript. As such, while the manuscript could be significantly improved by additional experiments, I think that most of the issues I observe can be addressed with alterations in the language to acknowledge the limitations of the data presented. Specific suggestions:

- 1) Abstract: Change to "These findings provide a potential mechanistic link between energy imbalance and C9-ALS/FTD pathogenesis and suggest a feedforward loop model that opens several opportunities for therapeutic intervention."
- 2) Line 296: In this study, we identified a novel link between energy imbalance and disease pathogenesis in models of C9orf72-ALS/FTD, which potentially reveals some opportunities for therapeutic intervention.
- 3) Lines 302-321: There is a lot of speculation on potential therapeutic interventions in patients here that are not warranted by the data in the manuscript. Recommending therapies or even clinical trials based on the data presented in this manuscript is premature and should be significantly limited.
- 4) Line 321: Overall, our findings warrant additional preclinical efforts to determine the impact of preserving bioenergetic homeostasis in C9orf72 ALS/FTD model systems - in both symptomatic and presymptomatic conditions in the hopes that such approaches might reveal potential therapies that can be tested in C9orf72 ALS/FTD patients.
- 5) Line 341: This is too speculative on ISRIB.
- 6) The authors should include a paragraph that addresses the limitations of the studies performed (see paragraph 2, this review) and provide future directions or approaches that might address these.

Response: We appreciate the positive comments on our work's overall interest and significance and the constructive suggestions for improvement. We acknowledge the observations regarding the limitations of our interpretation of some of our results and appreciate the suggestions for addressing these concerns. We have carefully considered the recommendations for language alterations to convey the scope and limitations of our findings. The point about the need for additional experiments to thoroughly test our hypothesis is well taken, and we agree that exploring the impact of preserving bioenergetic homeostasis in C9orf72 ALS/FTD in in-vivo model systems is an essential avenue for future research.

Based on the specific suggestions, we made the following changes (in red throughout the revised version of our manuscript):

- 1) **Abstract:** The abstract has been revised to reflect a more cautious interpretation of our findings and emphasize the potential mechanistic link between energy imbalance and C9-ALS/FTD pathogenesis without overemphasizing therapeutic opportunities.
- 2) **Line 296:** The text has been revised to align with the recommended cautious interpretation.
- 3) **Lines 302-321:** This section has been revised to limit speculation on potential therapeutic interventions, ensuring that our recommendations are more closely aligned with the data presented in the manuscript.
- 4) **Line 321:** We have acknowledged that our findings warrant additional preclinical efforts without prematurely recommending specific therapies or clinical trials.
- 5) **Line 341:** We tempered our speculation on ISRIB to align with a more cautious interpretation.
- 6) **Limitations and Future Directions:** We have included a paragraph addressing the limitations of our studies and providing insights into future directions and approaches that may address these limitations.

We genuinely appreciate the thoughtful feedback, and we believe that these revisions provided a more accurate representation of our findings.

Comments Referee #2: C9ORF72 hexanucleotide repeat expansion is the most common genetic cause of ALS and FTD. In this manuscript, the authors reported disruption of energy and amino acid production in the C9-BAC mice. They showed that the glucose hypometabolism leads to increased RAN translation of the repeat RNA by GCN2 kinase-mediated integrated stress response (ISR) using reporter cells. They further showed GCN2 inhibition can rescue the disease phenotypes of patient iPS-neurons, and glycolysis inhibition can increase poly-dipeptides and enhance phenotypes in C9-BAC mice. It has been reported that RAN translation can be increased by ISR. GCN2 and hypometabolism is one of the upstream factors that trigger ISR. It is interesting finding that this branch also contributes to DPR toxicity and disease phenotypes, using both in vivo and in vitro models. Some of the conclusions need more evidence to support before publication.

1. The activation of eIF2a pathway should be measured using phospho-eIF2a antibody. Only indirect evidence (such as ATF4) was used in the whole manuscript.

Response: In addition to quantifying ATF4 and downstream pathway effectors such as CHOP and GADD34, we have now shown the quantification of the phosphorylated eIF2a/eIF2a ratio and corresponding representative western blots in Figure 3e, e and Figure EV6f of the manuscript.

We are, instead, including an extended version of Fig. 3e here in this rebuttal for the reviewer's benefit. In this extended version of the figure, we have quantified, in addition to what we did for the figure presented in the main manuscript, the phosphor/dephospho ratio of iPS neurons that were treated with Na Arsenite. We used Na arsenite (500 uM) as a positive control to validate the phosphor/dephospho eIF2a antibodies and to maximize the activation in the iPS neurons of the ISR response. Our results showed that Na-arsenite, a well-known oxidative stressor activator of the ISR, increased the eIF2a phosphor/dephospho ratio by approximately 7-fold. However, we decided not to include this data set in the experimental design since

it is not directly relevant.

2. The effect on endogenous DPR levels need to be measured in patient derived neurons. Especially, the authors reasoned that the neuron survival changes is due to the DPR reduction. More data need to be provided to support this claim.

Response: see our response to point #4.

3. In Figure 5d, there is big variation of cell response to glucose deprivation. And the rescue effect is small in Figure 5e. More iPS cell lines and replicates are needed to support the conclusion.

Response: We have included an extra C9orf72 line in our experimental design to address the issue of cell response variation during glucose deprivation stress. To present a clear representation of the dataset, we have aggregated the results from three C9-iPSC lines in Fig. 5d and Fig. 5f. Furthermore, to ensure rigor, we have provided a breakdown of the individual cell line responses in Fig. EV5 a-d.

4. The DPR levels need to be measured using more quantitative methods, such as ELISA, in both cells and mice. The dot blot data are not very convincing.

Response: To address the reviewer's concern comprehensively, we have incorporated the SIMOA dataset into Figure 6 to provide additional support and complement the dot blot data, which has now been moved to Fig. EV6. Despite our best efforts, technical challenges hindered our attempts to measure poly(GP) levels in iPS-derived C9orf72 neurons using SIMOA. We have diligently engaged with Quanterix's technical support to address these issues, suspecting issues with reagent quality or machine functionality. However, despite extensive efforts, resolving these technical obstacles remains pending. Given this impasse, we have supplemented our analysis with dot blot quantification of poly(GP) formation in C9 lines at day 6, accompanied by appropriate controls, to reinforce our findings. While acknowledging that dot blot analysis may not match the accuracy of SIMOA, its robustness in detecting poly(GP) formation in various models and recent literature supports its use (<https://doi.org/10.1073/pnas.2123487119>; <https://www.nature.com/articles/s41593-021-00923-4>; <https://www.science.org/doi/epdf/10.1126/science.1232927>). We want to emphasize that we showcased our mechanism using the RAN-translation GA-GFP reporters, which are highly robust and reliable, enabling a comprehensive and accurate assessment of the pathway. However, the assessment of endogenous DPR production is still a technical challenge in most cases and may not provide clear conclusions. That is why we believed that the mechanism's demonstration could be established by using fluorescent RAN-translation reporters instead of, or in combination when feasible, with direct endogenous DPRs assessment.

In summary, while we continue to strive for the most rigorous quantification methods, including SIMOA, technical limitations have led us to rely on dot blot analysis, supplemented by appropriate controls, to support our findings. We are committed to resolving these technical challenges and will continue to explore alternative approaches to enhance the robustness of our data.

5. Overall, the statistics need to be improved. It is often not clear (or missing) the key information about replicates, cell numbers that were quantified.

Response: Essential information about statistical analysis was disclosed in the Material and Methods section of the original submission. However, we understand that this information might be missed if only reported in the Material and Methods section of the manuscript. We have therefore expanded the information section in the Figure legends to present statistical analysis and relative information. The newly added text in the revised manuscript is in red.

6. The authors should modify the writing and avoid overclaim. For example, the worsened phenotypes by 2DG treatment correlates with DPR change (which also need to be confirmed using better method). But this evidence itself does not prove the causal relationship. In particular, the timeline (which needs better description) of DPR and behavior changes seems not supporting DPR is the only cause.

Response: We acknowledge the significance of improving the language to prevent exaggeration. We are grateful for the chance to clarify certain aspects, which we have done in response to the same feedback from reviewer 1. We understand the concern about claiming a causal relationship regarding the correlation between worsened phenotypes with 2DG treatment and DPR changes. We have revised the text to emphasize the observed correlation while acknowledging the need for further investigation to establish causation. Please also see our response to point #4.

Dear Prof. Trotti,

Thank you for the submission of your revised manuscript. We have now received the enclosed reports from the referees. Referee 2 still has a few more suggestions that I would like you to address and incorporate before we can proceed with the official acceptance of your manuscript. Please co-submit with your final ms a point-by-point response to all last concerns.

A few editorial requests will also need to be addressed:

- Please reduce the number of keywords to 5.
- Please add full first author names to the ms file.
- Please remove the author credits from the ms file. All credits need to be entered during online ms submission into our online system.
- A completed author checklist needs to be uploaded with your final ms file. You can find the checklist here: <https://www.embopress.org/page/journal/14693178/authorguide>. The completed author checklist will also be part of our transparent peer-review file.
- Some funding info is missing in our online submission system, please add all funding info there and in the ms file.
- Please add the missing callout for Fig. 2f to the ms text.
- Table EV1 has 2 sheets and should be called Dataset EV1. Please correct the name, add a table title and legend to the excel file, and correct the callout in the ms text.
- The source data (SD) of each main figure need to be grouped into one folder and one SD folder per figure should be uploaded. SD for EV figures should be zipped into one folder.
- The manuscript sections should be in the following order: Title page - Abstract & Keywords - Introduction - Results - Discussion - Materials & Methods - Data Availability - Acknowledgments - Disclosure Statement & Competing Interests - References - Figure Legends - Tables with legends - Expanded View Figure Legends.
- We unfortunately can only process 5 EV figures. Can you may be try to rearrange the figures, or combine 2 ? If not, additional supplementary data need to uploaded as an Appendix file. Please see our guide to authors for more information.
- Please note that the box plots need to be defined in terms of minima, maxima, centre, bounds of box and whiskers, and percentile in the legends of figures 3e; 5e, g; 6f; EV 6e, g.
- For all figures with $n < 2$ no statistics should be calculated. Especially not when $n = 1$ and 3 technical replicates. Please remove the error bars and show all individual data points instead.
- In Fig 7e please specify if $n = 4$ refers to technical or biological replicates?
- Please note that the statistical test information for figure 4c is incorrectly mentioned as 4d in the data information section. This needs to be rectified.
- Please note that the legend for figure 5a is incorrectly labelled as 5d. This needs to be rectified.
- Please note that the figure panel EV 6g is not labelled in the figure. This needs to be rectified.
- Please note that the statistical test information for figure EV 6e, g is incorrectly labelled as EV 6e-f in the data information section. This needs to be rectified.
- Please define the annotated p values ****/* in the legend of figure EV 5a-d; as appropriate.
- Please indicate the statistical test used for data analysis in the legends of figures 3a; EV 2a; EV 5a-d; EV 6a.
- Please note that in figures 2e-f, h-i; 7c-e; EV 1b-e; there is a mismatch between the annotated p values in the figure legend and the annotated p values in the figure file that should be corrected.
- Please note that scale bar and its definition are missing for figure EV 7b.

I would like to suggest some changes to the abstract that needs to be written in present tense. Do you agree with this, and please add more information where I indicate below:

The most prevalent genetic cause of both amyotrophic lateral sclerosis and frontotemporal dementia is a (GGGGCC)_n nucleotide repeat expansion (NRE) occurring in the first intron of the C9orf72 gene (C9). Brain glucose hypometabolism is consistently observed in C9-NRE carriers, even at pre-symptomatic stages, but its role in disease pathogenesis is unknown. Here, we show alterations in glucose metabolic pathways and ATP levels in the brains of asymptomatic C9-BAC mice. We find that, through activation of the GCN2 kinase, glucose hypometabolism drives the production of dipeptide repeat proteins (DPRs), impairs the survival of C9 patient-derived neurons, and triggers motor dysfunction in C9-BAC mice. We also show that one of the arginine-rich DPRs (PR) can directly contribute to glucose metabolism and metabolic stress [It would be good to briefly explain how here]. Our findings provide a potential mechanistic link between energy imbalances and C9-ALS/FTD pathogenesis and suggest a feedforward loop model with potential opportunities for therapeutic intervention.

EMBO press papers are accompanied online by A) a short (1-2 sentences) summary of the findings and their significance, B) 2-3 bullet points highlighting key results and C) a synopsis image that is exactly 550 pixels wide and 200-600 pixels high (the height is variable). You can either show a model or key data in the synopsis image. Please note that text needs to be readable at the final size. Please send us this information along with the final manuscript.

Referee #1:

the reviewers have sufficiently addressed my concerns and have returned what I think is an excellent manuscript.

Referee #2:

The authors have improved the manuscript by adding more data and editing the writing. Here are the remaining concerns:
Figure EV6f: there is huge signal from the saline control. It is unclear how the authors quantified the gel. No increase of p-eIF2a in the 2DG samples from the gel.
For all the Kaplan-Meier survival analysis, it is better to include error bars to show the variations of technical/biological replicates. The information of the 3 control and 3 C9 patient-derived iPSC lines should be provided. The authors did not mention these are isogenic pairs. Then it is not clear how they were paired for comparison, control-A vs C9-A, etc? The 3 lines should be regarded as biological replicates, and the control group (n=3) should be compared with the C9 group (n=3) for the statistical comparison to determine whether there are significant differences.
In patient neurons, why DPR is not increased under glucose deprived condition?

Below is a point-by-point response to all of the last concerns raised by reviewer #2 and the editor.

Point-by-Point Rebuttal to the Editor

- Please reduce the number of keywords to 5. – *It has been reduced*
- Please add full first author names to the ms file. – *First author names have been spelled out.*
- Please remove the author credits from the ms file. All credits need to be entered during online ms submission into our online system. - *Done*
- A completed author checklist needs to be uploaded with your final ms file. You can find the checklist here: <https://www.embopress.org/page/journal/14693178/authorguide>; The completed author checklist will also be part of our transparent peer-review file. – *Done*
- Some funding info is missing in our online submission system, please add all funding info there and in the ms file. – *Done*
- Please add the missing callout for Fig. 2f to the ms text. - *Done*
- Table EV1 has 2 sheets and should be called Dataset EV1. Please correct the name, add a table title and legend to the excel file, and correct the callout in the ms text. - *Done*
- The source data (SD) of each main figure need to be grouped into one folder and one SD folder per figure should be uploaded. SD for EV figures should be zipped into one folder. – *This was done already for the previous submission.*
- The manuscript sections should be in the following order: Title page - Abstract & Keywords - Introduction - Results - Discussion - Materials & Methods - Data Availability - Acknowledgments - Disclosure Statement & Competing Interests - References - Figure Legends - Tables with legends - Expanded View Figure Legends. - *Done*
- We unfortunately can only process 5 EV figures. Can you may be try to rearrange the figures, or combine 2 ? If not, additional supplementary data need to be uploaded as an Appendix file. Please see our guide to authors for more information. – *Comment Retracted by the editor*
- Please note that the box plots need to be defined in terms of minima, maxima, centre, bounds of box and whiskers, and percentile in the legends of figures 3e; 5e, g; 6f; EV 6e, g. - *Done*
- For all figures with $n < 2$ no statistics should be calculated. Especially not when $n = 1$ and 3 technical replicates. Please remove the error bars and show all individual data points instead. - *Done*
- In Fig 7e please specify if $n = 4$ refers to technical or biological replicates? - *Done*
- Please note that the statistical test information for figure 4c is incorrectly mentioned as 4d in the data information section. This needs to be rectified. - *Done*
- Please note that the legend for figure 5a is incorrectly labelled as 5d. This needs to be rectified. - *Done*

- Please note that the figure panel EV 6g is not labelled in the figure. This needs to be rectified. - *Done*
- Please note that the statistical test information for figure EV 6e, g is incorrectly labelled as EV 6e-f in the data information section. This needs to be rectified. - *Done*
- Please define the annotated p values ****/* in the legend of figure EV 5a-d; as appropriate. - *Done*
- Please indicate the statistical test used for data analysis in the legends of figures 3a; EV 2a; EV 5a-d; EV 6a. *The statistical test applies is the Fischer's exact test with a Benjamini-Hochberg False Discovery Rate (FDR).*
- Please note that in figures 2e-f, h-i; 7c-e; EV 1b-e; there is a mismatch between the annotated p values in the figure legend and the annotated p values in the figure file that should be corrected. - *Done*
- Please note that scale bar and its definition are missing for figure EV 7b. - *Done*
- I would like to suggest some changes to the abstract that needs to be written in present tense. Do you agree with this, and please add more information where I indicate below: *Abstract changes accepted. More information has been added where indicated*

EMBO press papers are accompanied online by

- A) a short (1-2 sentences) summary of the findings and their significance.
Prolonged imbalances in glucose metabolism increase RAN translation and accumulation of DPRs, heightening neuronal vulnerability in in-vitro and in-vivo models of C9orf72-ALS/FTD.
- B) 2-3 bullet points highlighting key results
 - C9-BAC mice show dysregulated brain metabolites production.*
 - Glucose homeostasis is critical to maintaining RAN translation of the C9 repeats at low levels.*
 - Neurons bearing the ALS/FTD causative C9orf72 mutation are more vulnerable to glucose deficiencies than wild-type neurons.*
- C) a synopsis image that is exactly 550 pixels wide and 200-600 pixels high (the height is variable). – *Done shared with BioRender*

Point-by-Point Rebuttal to Referee #2

- Figure EV6f: There is a huge signal from the saline control. It is unclear how the authors quantified the gel. There is no increase of p-eIF2a in the 2DG samples from the gel. **Response:** We concur with the reviewer's observation that there is no significant increase in phospho-eIF2a. However, there is a reduction in the dephospho form of eIF2a, suggesting an increase in turnover. Although we did not present a quantification of eIF2a in the paper, we are including it here for the referee.

- For all the Kaplan-Meier survival analysis, it is better to include error bars to show the variations of technical/biological replicates. The information of the 3 control and 3 C9 patient-derived iPSC lines should be provided. The authors did not mention these are isogenic pairs. Then it is not clear how they were paired for comparison, control-A vs C9-A, etc? The 3 lines should be regarded as biological replicates, and the control group (n=3) should be compared with the C9 group (n=3) for the statistical comparison to determine whether there are significant differences. **Response:** We have added the bands indicating the 95% confidence interval (CI) to the K-M graphs. We apologize for not including the information that the iPSC lines are not isogenic. To our knowledge, there are no C9 isogenic lines available as of now due to the intrinsic technical difficulties and several factors:
 - 1. Genetic Instability.** The C9orf72 repeat expansion is highly unstable, characterized by somatic repeat length variation between different tissues and individuals. This instability poses a challenge in establishing stable iPSC lines with consistent repeat lengths, as the repeat expansions may undergo expansion or contraction during reprogramming or subsequent cell culture.
 - 2. Repeat Length Heterogeneity.** The repeat expansions in C9orf72-associated diseases can vary widely in length, ranging from tens to thousands of repeats. This heterogeneity makes it difficult to create isogenic iPSC lines with precisely matched repeat lengths, which is essential for studying the specific effects of repeat expansions on cellular phenotypes.
 - 3. Epigenetic Variability.** Epigenetic modifications, such as DNA methylation and histone acetylation, play crucial roles in regulating gene expression and cellular identity. However, these epigenetic marks can vary between iPSC lines derived from different individuals or even between different clones of the same individual, complicating efforts to establish truly isogenic iPSC lines.
 - 4. Differentiation Efficiency.** Even after successfully generating iPSC lines, differentiating them into disease-relevant cell types, such as motor neurons or cortical neurons, can be challenging and may exhibit variability between different lines or clones. This variability can obscure subtle phenotypic differences caused

by C9orf72 repeat expansions, making it difficult to draw meaningful conclusions from comparative studies.

We have included a table with all the necessary information regarding the control and C9orf72 iPSC lines used in this revised version of our manuscript.

We apologize for any confusion regarding the stats. We combined the three control lines and the three C9 lines for statistical comparison. In Figure EV5A, we represented the breakout of % survival of each individual iPSC line, and the comparison was performed between three replicates of one control line and three technical replicates of a C9 line. It is important to note that the lines compared against each other were originated by the same lab (i.e. Lines "A" from Dr. Ward's Lab, etc.) (see Table 4)

- In patient neurons, why DPR is not increased under glucose deprived condition?
Response: It's important to clarify that our experimental setup did not involve complete glucose deprivation but rather a nuanced condition comprising 15mM glucose supplemented with 10mM 2DG. This specific medium composition was chosen based on its ability to mimic glucose scarcity while maintaining some level of energy metabolizing activity. It's plausible that under these conditions, the cellular response could differ from what would be observed in a scenario of absolute glucose deprivation, potentially influencing the dynamics of DPR production. Notably, our findings showed a significant modulation of DPR production levels under 2DG conditions upon treatment with A92. This pharmacological intervention, targeting the GCN2-specific pathway/activity, effectively attenuated DPR levels. Such observations underscore the complexity of DPR regulation and hold promise for the development of targeted therapeutic strategies aimed at mitigating DPR-associated pathologies in neurodegenerative diseases. To summarize, it might appear confusing that there is no noticeable rise in DPRs during glucose-starved conditions. However, a careful examination of the complex biology governing DPR regulation can help clarify this phenomenon. Our research not only enhances our comprehension of DPR dynamics but also highlights the potential therapeutic benefits of focused interventions in ameliorating DPR-induced neurotoxicity.

Prof. Davide Trotti
Thomas Jefferson University
Neuroscience, Weinberg ALS center
900 Walnut street
Philadelphia, PA 19107
United States

Dear Prof. Trotti,

I am very pleased to accept your manuscript for publication in the next available issue of EMBO reports. Thank you for your contribution to our journal.

Yours sincerely,
